# Glutamate is required for depression but not potentiation of long-term presynaptic function

Zahid Padamsey[1,2]*, Rudi Tong[1], Nigel Emptage[1]*

[1]Department of Pharmacology, University of Oxford, Oxford, United Kingdom; [2]Centre for Discovery Brain Sciences, University of Edinburgh, Edinburgh, UK

**Abstract** Hebbian plasticity is thought to require glutamate signalling. We show this is not the case for hippocampal presynaptic long-term potentiation ($LTP_{pre}$), which is expressed as an increase in transmitter release probability ($P_r$). We find that $LTP_{pre}$ can be induced by pairing pre- and postsynaptic spiking in the absence of glutamate signalling. $LTP_{pre}$ induction involves a non-canonical mechanism of retrograde nitric oxide signalling, which is triggered by $Ca^{2+}$ influx from L-type voltage-gated $Ca^{2+}$ channels, not postsynaptic NMDA receptors (NMDARs), and does not require glutamate release. When glutamate release occurs, it decreases $P_r$ by activating presynaptic NMDARs, and promotes presynaptic long-term depression. Net changes in $P_r$, therefore, depend on two opposing factors: (1) Hebbian activity, which increases $P_r$, and (2) glutamate release, which decreases $P_r$. Accordingly, release failures during Hebbian activity promote $LTP_{pre}$ induction. Our findings reveal a novel framework of presynaptic plasticity that radically differs from traditional models of postsynaptic plasticity.

DOI: https://doi.org/10.7554/eLife.29688.001

*For correspondence:
zahid.padamsey@ed.ac.uk (ZP);
nigel.emptage@pharm.ox.ac.uk
(NE)

**Competing interests:** The authors declare that no competing interests exist.

## Introduction

Learning and memory are thought to require synaptic plasticity, which refers to the capacity for synaptic connections in the brain to change with experience. The most frequently studied forms of synaptic plasticity are long-term potentiation (LTP) and long-term depression (LTD), which respectively involve long-lasting increases and decreases in synaptic transmission. LTP and LTD can be expressed either postsynaptically ($LTP_{post}$ or $LTD_{post}$), as changes in AMPA receptor (AMPAR) number, or presynaptically ($LTP_{pre}$ or $LTD_{pre}$), as changes in glutamate release probability ($P_r$) (*Padamsey and Emptage, 2011*; *Bliss and Collingridge, 2013*; *Larkman and Jack, 1995*; *Lisman, 2003*; *Lisman and Raghavachari, 2006*; *Padamsey and Emptage, 2014*). Traditionally, postsynaptic NMDA receptor (NMDAR) activation is believed to be important for both pre- and postsynaptic forms of plasticity (*Bliss and Collingridge, 2013*; *Lüscher and Malenka, 2012*). Postsynaptic changes, in particular, have been causally and convincingly linked to NMDAR-dependent $Ca^{2+}$ influx, which, via the activation of postsynaptic $Ca^{2+}$-sensitive kinases and phosphatases, triggers changes in the number of synaptic AMPARs (*Lüscher and Malenka, 2012*). The link between postsynaptic NMDAR activation and presynaptic plasticity, however, is less clear. In the case of $LTP_{pre}$ induction, it is traditionally thought that $Ca^{2+}$ influx through postsynaptic NMDARs triggers the synthesis and release of a retrograde signal, most likely nitric oxide (NO), which in turn triggers increases in $P_r$ (*Padamsey and Emptage, 2014*; *Garthwaite and Boulton, 1995*) (though other forms of presynaptic plasticity exist [*Yang and Calakos, 2013*; *Castillo, 2012*]). Consistent with this, several studies have demonstrated that $LTP_{pre}$ induction is impaired by the blockade of NMDAR or NO signalling (*Ryan et al., 1996*; *Ratnayaka et al., 2012*; *Emptage et al., 2003*; *Bliss and Collingridge, 2013*; *Enoki et al., 2009*; *Nikonenko et al., 2003*; *Stanton et al., 2005*; *Padamsey and Emptage, 2014*;

**eLife digest** Neurons communicate with one another at junctions called synapses. One neuron at the synapse releases a chemical substance called a neurotransmitter, which binds to and activates the other neuron. The release of neurotransmitter thus enables the electrical activity of one cell to influence the electrical activity of another. The efficiency of this communication can change over time, as is thought to occur during learning. If the neurons on both sides of a synapse are repeatedly active at the same time, the ability of the neurons to transmit electrical signals to each other increases.

One way that communication between neurons can become more efficient is if the first neuron becomes more likely to release neurotransmitter. Most synapses in the brain release a neurotransmitter called glutamate, and most types of learning involve changes in the efficiency of communication at glutamatergic synapses. But glutamate release is unreliable. Active glutamatergic neurons fail to release glutamate about 80% of the time. If glutamate has a key role in learning, how does the brain learn efficiently when glutamate release is so unlikely?

To find out, Padamsey et al. studied glutamatergic synapses in slices of tissue from mouse and rat brains. When both neurons at a synapse were repeatedly active at the same time, the first neuron would sometimes become more likely to release glutamate. But this only happened at synapses in which the first neuron usually failed to release glutamate in the first place. This suggests that communication failures help to drive change at synapses. When two neurons that are often active at the same time do not communicate efficiently, this failure triggers molecular changes that make future communication more reliable.

Previous results have shown that synapses can change when glutamate release occurs. The current results show that they can also change when it does not. This means that the brain can continue to learn despite frequent communication failures between neurons. Many neurological disorders, including Alzheimer's disease, show altered glutamate signalling at synapses. Padamsey et al. hope that a better understanding of this process will lead to new therapies for these disorders.
DOI: https://doi.org/10.7554/eLife.29688.002

*Johnstone and Raymond, 2011*). However, some groups have found that presynaptic enhancement can be induced in the presence of NMDAR antagonists (*Blundon and Zakharenko, 2008*; *Zakharenko et al., 2003*; *Bayazitov et al., 2007*; *Zakharenko et al., 2001*; *Stricker et al., 1999*) (but see [*Grover and Yan, 1999a*; *Grover, 1998*]). Under these conditions, presynaptic plasticity relies on L-type voltage-gated $Ca^{2+}$ channel (L-VGCC) activation (*Blundon and Zakharenko, 2008*; *Zakharenko et al., 2003*; *Bayazitov et al., 2007*; *Zakharenko et al., 2001*), but may still depend on NO signalling (*Padamsey and Emptage, 2014*; *Pigott and Garthwaite, 2016*). These findings suggest that $LTP_{pre}$ may require neither the activation of postsynaptic NMDARs nor NMDAR-dependent NO synthesis; nonetheless, results across studies are largely inconsistent (*Padamsey and Emptage, 2014*), and the exact mechanism and retrograde signal underlying $LTP_{pre}$ induction remains unclear.

The role of glutamate signalling in presynaptic plasticity is also unclear. Glutamate release is of course necessary to drive the postsynaptic depolarization required for the induction of both $LTP_{pre}$ and $LTP_{post}$. However, this does not explain why any given synapse needs to release glutamate in order to be potentiated, since depolarization triggered by one synapse can affect another, either directly via electrotonic spread, or indirectly via the actions of dendritic or somatic spikes. The necessity for site-specific glutamate release in LTP induction, at least in the case of $LTP_{post}$, is instead imposed by the strict requirement of postsynaptic NMDAR-mediated $Ca^{2+}$ influx for potentiation (*Bliss and Collingridge, 2013*; *Lüscher and Malenka, 2012*). However, that NMDARs may not to be necessary for the induction of $LTP_{pre}$ (*Blundon and Zakharenko, 2008*; *Padamsey and Emptage, 2014*) suggests that the role of synapse-specific glutamate release in presynaptic plasticity may be different. Indeed, a common finding across a number of studies is that high $P_r$ synapses are more likely to show $LTD_{pre}$, whereas low $P_r$ synapses are more likely to show $LTP_{pre}$ (*Ryan et al., 1996*; *Slutsky et al., 2004*; *Larkman et al., 1992*; *Hardingham et al., 2007*; *Sáez and Friedlander, 2009*). Moreover, glutamate release can induce $LTD_{pre}$ by acting on presynaptic NMDARs (*McGuinness et al., 2010*; *Rodríguez-Moreno et al., 2013*), or metabotropic glutamate receptors

(mGluRs) in the case of younger tissue (*Zakharenko et al., 2002*). Thus, enhanced glutamate release at a presynaptic terminal, unlike at a dendritic spine (*Lüscher and Malenka, 2012*; *Harvey and Svoboda, 2007*; *Matsuzaki et al., 2004*; *Makino and Malinow, 2009*), may not necessarily result in enhanced potentiation, but instead promote depression. Several studies have also demonstrated that presynaptic terminals initially releasing little or no glutamate are reliably potentiated following tetanic stimulation (*Ryan et al., 1996*; *Emptage et al., 2003*; *Slutsky et al., 2004*; *Larkman et al., 1992*; *Hardingham et al., 2007*; *Sáez and Friedlander, 2009*; *McGuinness et al., 2010*; *Enoki et al., 2009*). How low $P_r$ synapses, including those that are putatively silent, can undergo activity-dependent potentiation raises questions as to the necessity of synapse-specific glutamate release in presynaptic plasticity.

Here we re-examined the mechanisms underlying activity-dependent presynaptic changes at CA3-CA1 hippocampal synapses, with a particular focus on understanding the role of glutamate in presynaptic plasticity. We find that, contrary to current thinking, Hebbian activity, via L-VGCC-triggered NO signalling, is sufficient to induce $LTP_{pre}$ without the need for synapse-specific signalling by glutamate. When glutamate release occurs, it inhibits $LTP_{pre}$ and instead promotes $LTD_{pre}$ by activating presynaptic NMDARs. Thus, for presynaptic potentiation to occur, a presynaptic neuron must not only fire together with its postsynaptic partner, but it must also fail to release glutamate. Our findings reveal a novel set of rules and mechanisms governing presynaptic plasticity that are distinct from those associated with traditional, postsynaptic models of plasticity.

## Results

### High frequency presynaptic activity inhibits $LTP_{pre}$ and promotes $LTD_{pre}$

We started by examining how manipulating glutamatergic signalling at synapses would affect activity-driven changes in presynaptic function. We recorded excitatory postsynaptic potentials (EPSPs) in CA1 neurons in cultured hippocampal slices. Cells were recorded using patch electrodes (4–8 MΩ) and EPSPs were evoked by Schaffer-collateral stimulation. Baseline EPSP recordings were kept short (5 min) to minimize dialysis as we found that longer baseline recordings prevented $LTP_{pre}$ induction (see Materials and methods). For LTP induction we used a pairing protocol, in which individual presynaptic stimuli were causally paired with postsynaptic spiking, 60 times at 5 Hz. Because $LTP_{pre}$ is preferentially induced under conditions of strong postsynaptic depolarization (*Padamsey and Emptage, 2014*; *Blundon and Zakharenko, 2008*; *Zakharenko et al., 2003*; *Bayazitov et al., 2007*; *Zakharenko et al., 2001*; *Stricker et al., 1999*), we paired presynaptic stimuli with a current injection of sufficient amplitude to generate 3–6 postsynaptic spikes over a 50–60 ms time course. Spikes tended to broaden over the time course of injection, and the resulting waveform resembled a complex spike (*Figure 1A*), which is known to efficiently drive LTP *in vitro* (*Thomas et al., 1998*; *Remy and Spruston, 2007*; *Golding et al., 2002*; *Hardie and Spruston, 2009*), and has been recorded in the hippocampus *in vivo* (*Ranck, 1973*; *Grienberger et al., 2014*). We found that this pairing protocol produced robust and reliable LTP (fold $\Delta EPSP_{slope}$: 1.88 ± 0.24; n = 6 cells; vs 1.0: p<0.05; *Figure 1B,C*), which had a presynaptic component of expression, as assessed by a decrease in the paired pulse ratio (PPR) ($\Delta PPR$: −0.39 ± 0.15; n = 6 cells; vs 0: p<0.05; *Figure 1D*). Changes in PPR were evident across a range of intervals; however, we chose to measure PPR at an interval of 70 ms, at which plasticity-induced changes tended to be maximal (*Figure 1—figure supplement 1A*).

We then examined the effects of elevating glutamate release during $LTP_{pre}$ induction. One physiological means of transiently increasing glutamate release probability ($P_r$) is to elevate the frequency of presynaptic activity (*Dobrunz and Stevens, 1997*; *Dobrunz and Stevens, 1997*). We therefore repeated our LTP experiments, but during induction, in the place of single presynaptic pulses, we used short, high frequency bursts of presynaptic stimuli to increase $P_r$ (*Figure 1—figure supplement 2*). The burst consisted of two pulses, delivered 5 ms apart, and resembled high-frequency bursting activity recorded in CA3 neurons *in vivo* (*Kowalski et al., 2016*). Remarkably, we found that pairing burst stimulation with postsynaptic depolarization produced significantly less LTP compared to single pulse pairings (fold $\Delta EPSP_{slope}$: 1.36 ± 0.13; n = 6 cells; vs. single pulse pairings: p<0.05), and was accompanied by no significant changes in PPR ($\Delta PPR$: 0.00 ± 0.04; n = 6 cells; vs 0: p=0.84; vs. single pulse pairings: p<0.01; *Figure 1D*). These findings suggest that high frequency presynaptic activity inhibits the induction of $LTP_{pre}$.

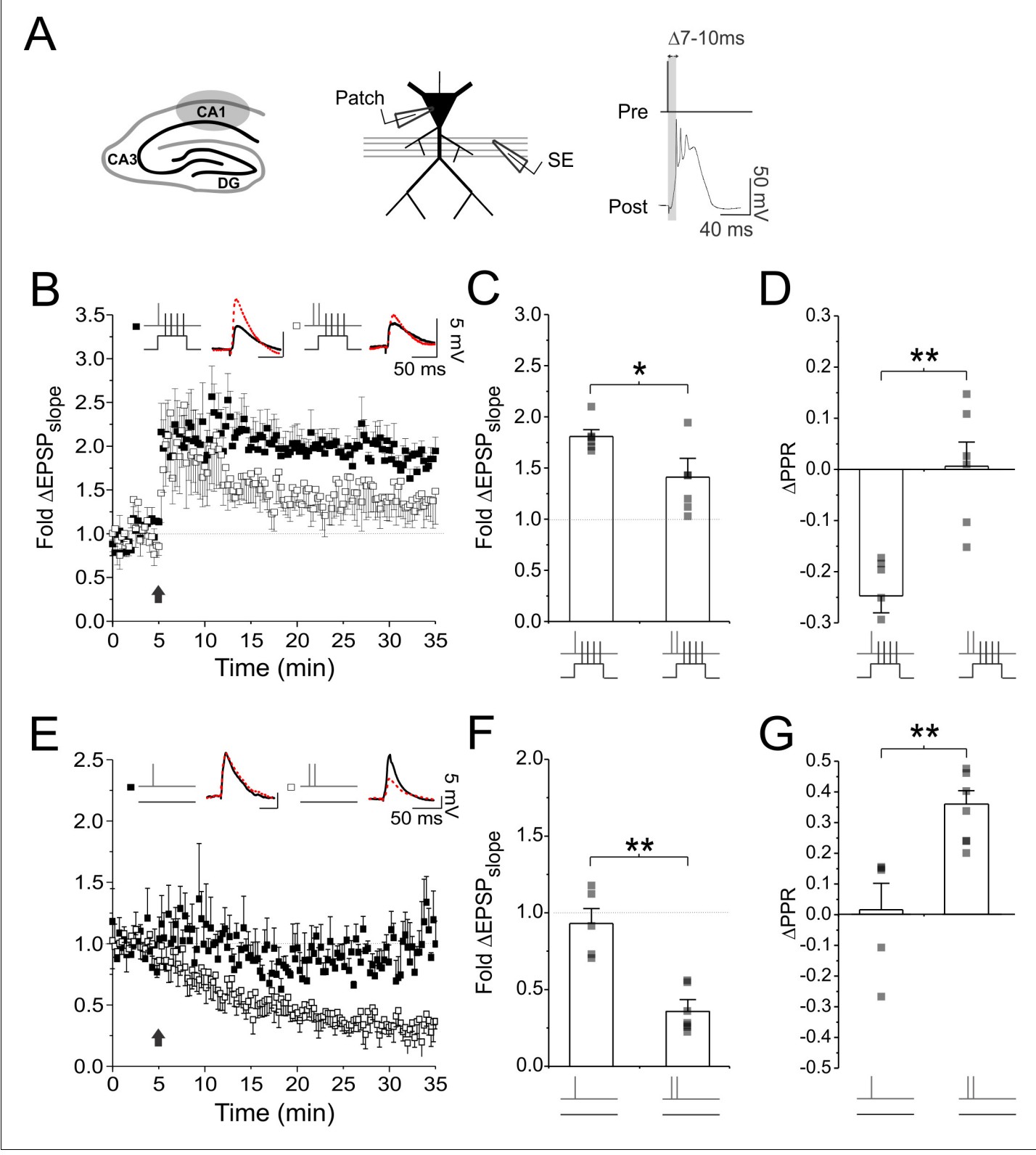

**Figure 1.** High frequency presynaptic activity inhibits LTP_pre and promotes LTD_pre. (**A**) Experimental setup. CA1 pyramidal neurons were recorded using whole-cell patch electrodes. LTP was induced by causal pairing of presynaptic activity with postsynaptic depolarization in the form of a complex spike. (**B,E**) Average fold changes of EPSP slope plotted against time. Following baseline recording, 60 presynaptic stimuli were delivered at 5 Hz, either in the presence (paired stimulation) or absence (unpaired stimulation) of postsynaptic depolarization. Presynaptic stimuli were delivered either as single

*Figure 1 continued*

pulses, or high frequency presynaptic bursts consisting of two pulses delivered 5 ms apart. Sample EPSP traces at baseline (black trace) and 30 min after plasticity induction (broken red trace) are shown as an inset above each graph. Average changes in (C,F) EPSP slope and (D,G) PPR measured 25–30 min after plasticity induction. High frequency bursts generated significantly less LTP$_{pre}$ with paired stimulation, and more LTD$_{pre}$ with unpaired stimulation, than did single pulse stimulation. Error bars represent S.E.M. (n = 5–8 cells per condition). Asterisks denotes significance differences between groups (*p<0.05; **p<0.01; Mann-Whitney test).

DOI: https://doi.org/10.7554/eLife.29688.003

The following figure supplements are available for figure 1:

**Figure supplement 1.** LTP$_{pre}$ and LTD$_{pre}$ induction are associated with changes in paired pulse ratio (PPR) at a range of paired pulse intervals.

DOI: https://doi.org/10.7554/eLife.29688.004

**Figure supplement 2.** Presynaptic burst stimulation increases P$_r$.

DOI: https://doi.org/10.7554/eLife.29688.005

To examine the effects of presynaptic stimulation alone, we repeated our experiments, but during LTP induction we omitted postsynaptic depolarization (unpaired stimulation). Under these conditions, when single presynaptic stimuli were delivered, we observed no significant change in the EPSP (fold $\Delta$EPSP$_{slope}$: 0.93 $\pm$ 0.10; n = 5 cells; vs 1.0: p=0.62; *Figure 1E,F*) or PPR (fold $\Delta$PPR: 0.01 $\pm$ 0.09; n = 5 cells; vs 0: p=0.81; *Figure 1F*). However, when high frequency bursts were delivered during induction, we observed a robust decrease in the EPSP (fold $\Delta$EPSP$_{slope}$: 0.42 $\pm$ 0.08; n = 8 cells; vs. single pairing: p<0.01; *Figure 1E,F*) and an increase in PPR (fold $\Delta$PPR: 0.36 $\pm$ 0.04; n = 8 cells; vs. single pulse pairing: p<0.01; *Figure 1G*; *Figure 1—figure supplement 1B*), suggesting that we had induced LTD with a presynaptic component of expression. Collectively, these findings demonstrate that high frequency presynaptic stimulation not only inhibits the induction of LTP$_{pre}$, but also promotes the induction of LTD$_{pre}$.

## Glutamate photolysis inhibits LTP$_{pre}$ and promotes LTD$_{pre}$

We next tested whether the effects of high frequency presynaptic stimulation on LTP$_{pre}$ and LTD$_{pre}$ were in fact due to synapses releasing glutamate more reliably, as opposed to other effects, such as an increase in Ca$^{2+}$ influx at the presynaptic terminal. To do so, we used glutamate uncaging instead of high-frequency presynaptic stimulation to artificially elevate P$_r$ at synapses during LTP induction. Glutamate uncaging was restricted to single synapses, and activity-dependent changes in presynaptic function (i.e. P$_r$) were assessed by imaging postsynaptic Ca$^{2+}$ transients (*Emptage et al., 2003*; *Emptage et al., 1999*). This technique relies on the fact that at most CA3-CA1 synapses single quanta of glutamate, through AMPAR-mediated depolarization, generate sufficient Ca$^{2+}$ influx from NMDAR and voltage-gated Ca$^{2+}$ channels (VGCCs) to be detected by Ca$^{2+}$-sensitive dyes (*Padamsey and Emptage, 2011*; *Grunditz et al., 2008*; *Emptage et al., 1999*). Consequently, the proportion of trials in which single presynaptic stimuli generate postsynaptic Ca$^{2+}$ transients can be used to calculate P$_r$ at single synapses (*Emptage et al., 1999*). Notably, estimates of P$_r$ measured at resting membrane potential are resilient to large perturbations of postsynaptic Ca$^{2+}$ influx (*Figure 2—figure supplement 1*; also see *Figure 6—figure supplement 1A–C*).

CA1 pyramidal neurons were filled with the Ca$^{2+}$-sensitive dye Oregon Green BAPTA-1 and a fluorescently-coated glass electrode was used to stimulate Schaffer-collaterals in the vicinity of an imaged dendrite (*Figure 2A*). Dendritic spines were sequentially scanned in order to identify those that were responsive to stimulation. To increase the likelihood of visually identifying responsive synapses, especially those with low basal release probabilities, we delivered two presynaptic stimuli, 70 ms apart, to transiently increase P$_r$ (*Figure 2B*). When a synapse was found that responded to stimulation, it always responded in an all-or-none manner, with Ca$^{2+}$ transients largely restricted to the spine head. As expected, Ca$^{2+}$ transients were more likely to be elicited by the second of the two presynaptic stimuli because of the effects of short-term facilitation. P$_r$ was calculated as the proportion of trials in which the first of the two presynaptic stimuli generated a fluorescent increase in the spine head; the second of the two presynaptic stimuli was ignored. Because of the additional time required to measure P$_r$, most of our imaging experiments were done in the absence of electrophysiological recordings in cells bolus-loaded with Ca$^{2+}$ sensitive dye; cells were only transiently patched for plasticity induction (see Materials and methods).

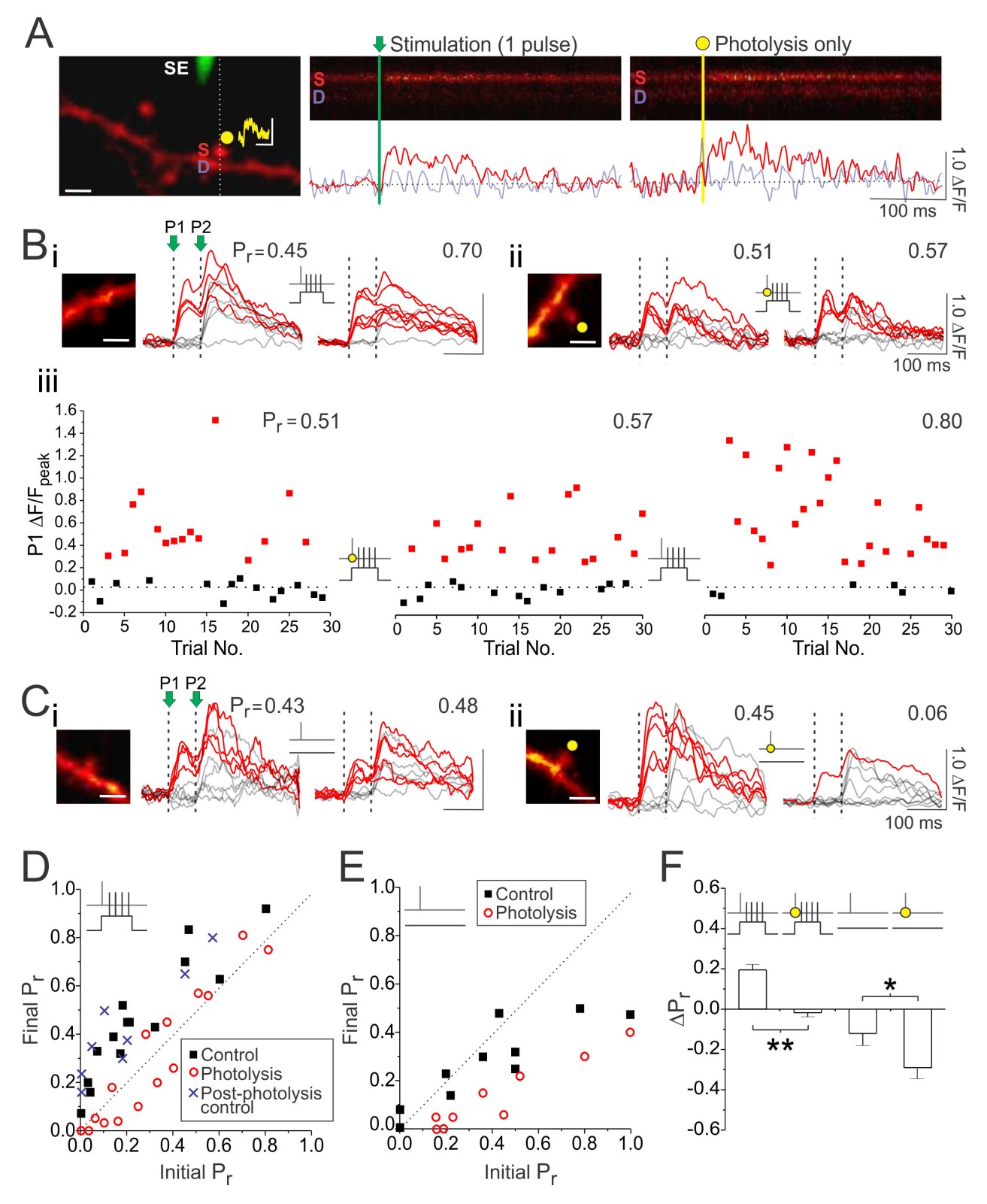

**Figure 2.** Glutamate photolysis inhibits LTP_pre and promotes LTD_pre. (**A**) Left, an image of a CA1 neuronal dendrite loaded with Oregon Green BAPTA-1 $Ca^{2+}$-sensitive dye. A stimulating electrode (SE; green) was placed close to the dendrite in order to activate spines within the vicinity (scale bar: 2 μm). A spine responsive to stimulation was located and then targeted for glutamate photolysis (yellow spot). An example of an uncaging-evoked synaptic potential is shown above the imaged spine (scale bar: 1 mV by 100 ms). During stimulation and photolysis, evoked $Ca^{2+}$ transients were rapidly imaged

*Figure 2 continued on next page*

*Figure 2 continued*

by restricting laser scanning to a line across the spine head and underlying dendrite (broken line). Right, samples of these $Ca^{2+}$ transients in both the spine (labelled S) and dendritic (labelled D) are shown. Below each line scan image are traces quantifying the fluorescence change (ΔF/F) for the spine (red trace; raw) and dendrite (purple trace; raw). Photolysis laser power was adjusted to elicit spine $Ca^{2+}$ transients comparable to that induced by single presynaptic stimuli. (Bi and Bii) Samples of 10 superimposed $Ca^{2+}$ traces evoked in imaged spines (white scale bar: 2 µm) by paired pulse stimulation (P1 and P2 were delivered 70 ms apart and are represented by vertical broken lines); red traces depict successful release events to the first (P1) of the two pulses. $Ca^{2+}$ traces are shown during baseline and 25–30 min following paired stimulation, delivered in either the (Bi) absence or (Bii) presence of glutamate photolysis (yellow circle). $P_r$ was calculated as the proportion of total stimulation trials in which the first pulse (P1) resulted in a successful release event. (Biii). For the experiment in (Bii), the peak ΔF/F in the spine head is plotted for the first pulse (P1) across 30 stimulation trials given at baseline, 30 min following paired stimulation with photolysis, and 30 min following a subsequent round of paired stimulation in the absence of photolysis. Red squares denote fluorescent increases above noise. Increases in $P_r$ could only be induced by paired stimulation delivered in the absence of glutamate photolysis. (Ci and Cii) As in (B) except for unpaired stimulation (60 pulses at 5 Hz), which alone had a negligible effect on low $P_r$ synapses ($P_r$ <0.5). Glutamate photolysis promoted greater decreases in $P_r$ than unpaired stimulation alone. (D,E) The final $P_r$ measured 25–30 min following stimulation is plotted against the initial $P_r$ measured at baseline for each imaged synapse. The broken diagonal line represents the expected trend if $P_r$ is unchanged by stimulation. The post-photolysis control group consisted of 8 synapses from the photolysis group that underwent a second round of paired stimulation, but in the absence of glutamate photolysis (see Biii for example). (F) Average changes in $P_r$. Error bars represent S.E.M. (n = 9–14 spines per condition). Asterisks denotes significance differences between groups (*p<0.05; **p<0.01; Mann-Whitney test).

DOI: https://doi.org/10.7554/eLife.29688.006

The following figure supplement is available for figure 2:

**Figure supplement 1.** Augmenting spine $Ca^{2+}$ signalling does not alter $P_r$ estimates.
DOI: https://doi.org/10.7554/eLife.29688.007

Consistent with our electrophysiological results, pairing single presynaptic stimuli with postsynaptic complex spikes (60 pairing at 5 Hz) evoked an increase in $P_r$ (ΔP$_r$: 0.19 ± 0.03; n = 14 spines; vs. 0: p<0.01; *Figure 2B,D,F*). We then repeated the experiment but this time, during LTP induction, each presynaptic stimulus was coupled with photolysis of caged glutamate at the synapse, regardless of whether the synapse released glutamate or not, in order to artificially elevate $P_r$ during stimulation. We adjusted the laser power to ensure that photolysis mimicked the fluorescent changes elicited by uniquantal glutamate release evoked by single presynaptic stimuli (ΔF/F; photolysis vs. stimulation: 0.46 ± 0.07 vs. 0.55 ± 0.09; n = 15 spines; p=0.23; *Figure 2A*). Remarkably, under these conditions, increases in $P_r$ at the target synapse were effectively abolished (ΔP$_r$: −0.02 ± 0.02; n = 15 spines; photolysis vs. control: p<0.01; *Figure 2B,D,F*). This demonstrates that, consistent with our hypothesis, transiently elevating glutamate signalling at synapses inhibited the induction of LTP$_{pre}$. In eight of the synapses imaged under these conditions, LTP induction was repeated for a second time, but in the absence of caged glutamate during photolysis; in these experiments, the expected increase in $P_r$ was observed (ΔP$_r$: 0.23 ± 0.02; n = 8 spines; vs. control: p=0.48; *Figure 2Biii*, post-photolysis control in *Figure 2D*). Increases in $P_r$ were also observed in a subset of control experiments, in which LTP induction was conducted in the presence of caged glutamate, but in the absence of photolytic laser exposure (ΔP$_r$: 0.25 ± 0.03; n = 8 spines; vs. control: p=0.15). These results suggest that the inhibitory effect of photolysis on $P_r$ was due to glutamate release, as opposed to non-specific effects of uncaging.

We also examined the effects of glutamate photolysis delivered in the absence of postsynaptic depolarization (unpaired stimulation). Delivery of 60 presynaptic stimuli at 5 Hz, consistent with our electrophysiological recordings, produced no changes in $P_r$ at the majority of synapses imaged (*Figure 2C,E,F*). We did, however, notice that synapses with initially high release probabilities ($P_r$ >0.5), showed a modest decrease in $P_r$ following unpaired stimulation (*Figure 2E*); this decrease was not likely to be detected by electrophysiological recordings because high $P_r$ synapses comprise an estimated <10% of synapses in our preparation (*Ward et al., 2006*). Remarkably, when we coupled each presynaptic stimulus with glutamate photolysis, we now observed decreases in $P_r$ at all imaged synapses, regardless of their initial $P_r$ (ΔP$_r$ photolysis vs. control: −0.33 ± 0.08 vs. −0.12 ± 0.06; n = 9,10 spines p<0.05; *Figure 2C,E,F*). These findings suggest that elevated glutamate release decreases $P_r$, and does so regardless of the level of postsynaptic depolarization that accompanies presynaptic activity; $P_r$ changes induced by paired or unpaired stimulation were always more negative compared to controls when glutamate signalling was augmented.

## LTP$_{pre}$ can be induced in full glutamate receptor blockade

Given that transiently elevating glutamate release probability, either by presynaptic bursts or glutamate photolysis, inhibited the induction of LTP$_{pre}$, we asked if glutamate signalling was required at all for driving increases in P$_r$ during paired stimulation, as traditionally believed. Physiologically, glutamate is clearly necessary for driving the postsynaptic spiking required for LTP, and all major classes of glutamate receptors including: AMPARs, Kainate receptors (KARs), NMDARs, and mGluRs can contribute to membrane depolarization (*Grienberger et al., 2014*; *Schiller and Schiller, 2001*; *Grover and Yan, 1999b*; *Chemin et al., 2003*). If, however, membrane depolarization is the only function of glutamate in LTP$_{pre}$ induction, then a presynaptic terminal could in principle be potentiated even if it failed to release glutamate, provided that its activity coincided with strong postsynaptic depolarization, as driven by glutamate release at other co-active synapses. If this is the case, we reasoned that we should be able to experimentally trigger LTP$_{pre}$ in a full glutamate receptor blockade provided that, during presynaptic stimulation, we supplemented the depolarizing effects of glutamate with somatic current injection. If, however, glutamate is additionally required for some form of synapse-specific signalling, as in the case of LTP$_{post}$ induction, then the induction of LTP$_{pre}$ should not be possible in full glutamate receptor blockade no matter how much we depolarize the neuron during presynaptic stimulation.

To test this possibility we attempted to induce LTP$_{pre}$ at CA3-CA1 synapses with all known glutamate receptors (AMPARs, KARs, NMDARs, and mGluRs) pharmacologically inhibited (10 µM NBQX, 100 µM D-AP5, 0.5 mM R,S-MCPG, 100 µM LY341495); we used AP5 instead of MK-801 in order to block both ionotropic and metabotropic effects associated with NMDAR activation (*Nabavi et al., 2013*). Given the additional time required for these experiments, we recorded from CA1 neurons using high-resistance patch electrodes (18–25 MΩ) to limit the effects of postsynaptic dialysis. Following pharmacological abolishment of the EPSP, we delivered paired stimulation as before, during which strong postsynaptic depolarization again took the form of a complex spike induced by somatic current injection (*Figure 3A*). The antagonist cocktail was then washed out in order to recover the EPSP. As expected with the use of high concentrations of NBQX (*Holbro et al., 2010*), EPSP recovery was never complete and varied across experiments (*Figure 3B,C*), and so it was necessary to compare the EPSP recorded from the pathway receiving paired stimulation to a second, independent control pathway recorded simultaneously (*Figure 3A,B*). We found that paired stimulation induced a robust enhancement of the EPSP in the stimulated pathway relative to the control pathway (fold ΔEPSP$_{slope}$; paired vs. control: 1.12 ± 0.13 vs. 0.71 ± 0.12; n = 7 cells; p<0.05; *Figure 3B, D*); this enhancement was not seen when pairings were anti-causal, with presynaptic stimuli following postsynaptic spiking (*Figure 3—figure supplement 1* ). Causal pairings resulted in a 1.72 ± 0.21 fold potentiation, which we estimated by normalizing the fold change in the EPSP of the paired pathway to that of the control pathway. Notably, EPSP recovery of the control pathway was not significantly different from experiments in which drugs were applied in the absence of paired stimulation (control vs. drugs-only: 0.71 ± 0.12 vs. 0.54 ± 0.11; n = 7, 5 cells; p=0.59; *Figure 3C,D*), suggesting that LTP was restricted to only synapses that were active during the pairing. LTP was also associated with a significant decrease in PPR (paired vs. control ΔPPR: −0.28 ± 0.06 vs. 0.03 ± 0.03; n = 6 cells; p<0.05; *Figure 3E*), that again, was only found in the paired pathway, suggesting that LTP induction was both presynaptic and site-specific. Similar site-specific enhancements in presynaptic function could be induced under full glutamate receptor blockade in acute hippocampal slices (*Figure 3—figure supplement 2*).

In contrast to these findings, several studies have demonstrated that NMDAR blockade alone impairs LTP induction, even presynaptically (*Ryan et al., 1996*; *Ratnayaka, 2012*; *Emptage et al., 2003*; *Bliss and Collingridge, 2013*; *Enoki et al., 2009*; *Nikonenko et al., 2003*; *Stanton et al., 2005*; *Padamsey and Emptage, 2014*). However, it is important to recognize that NMDARs, like all glutamate receptors, can contribute to postsynaptic depolarization. The NMDARs are particularly potent sources of depolarization, especially given their role in dendritic (*Schiller and Schiller, 2001*; *Losonczy and Magee, 2006*) and somatic spiking (*Grienberger et al., 2014*). Thus, it is possible that NMDAR blockade inhibits LTP$_{pre}$ expression by inhibiting postsynaptic depolarization. This is less likely to be an issue when strong postsynaptic depolarization is driven via somatic current injection, as in our experiment (*Figure 3*), than when depolarization is driven by presynaptic stimulation alone (e.g. tetanic stimulation). To test this reasoning, we induced LTP using standard 100 Hz tetanic

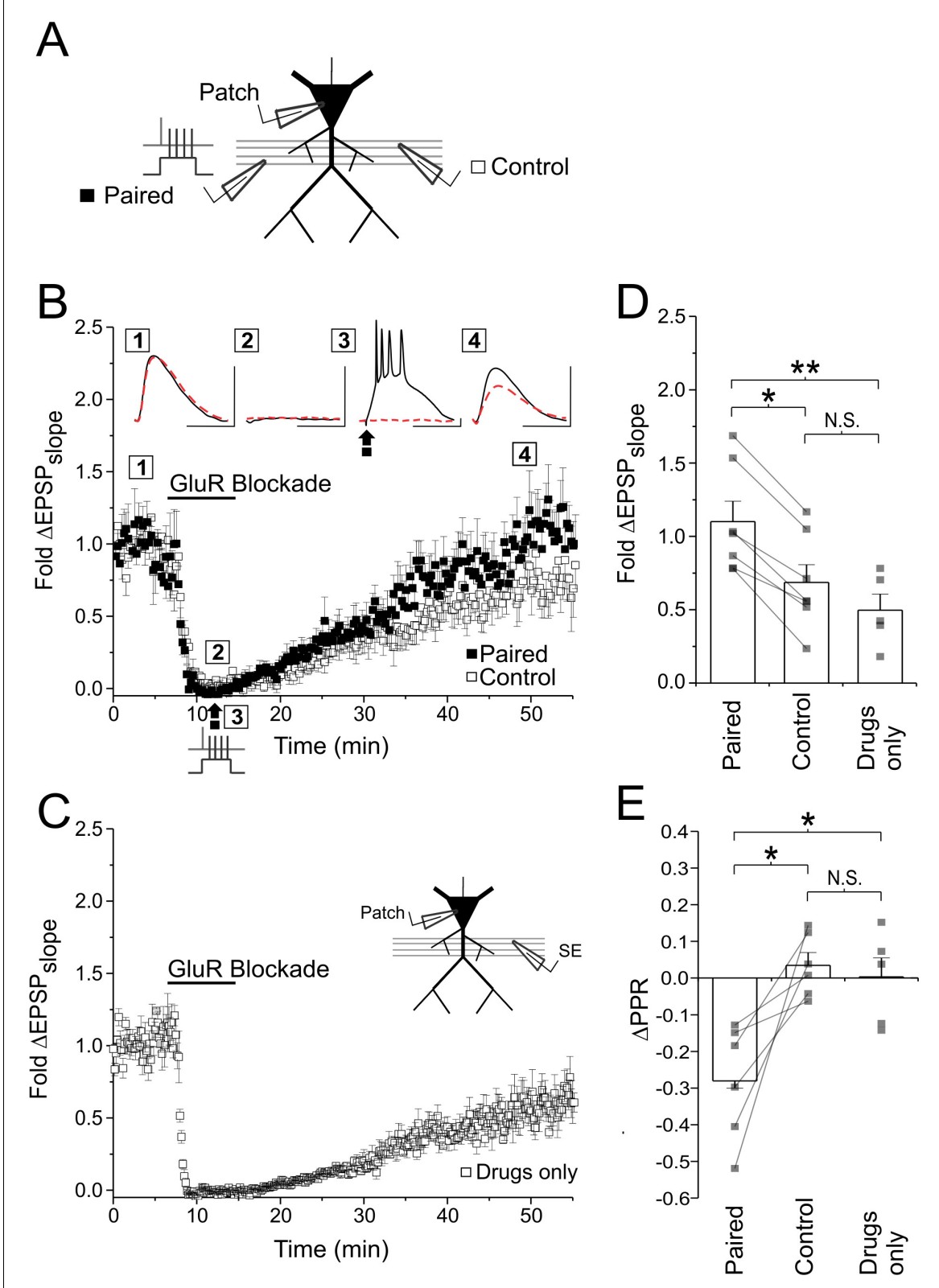

**Figure 3.** Induction of LTP$_{pre}$ in glutamate receptor blockade. (**A**) Experimental setup. EPSPs were recorded from two independent Schaffer-collateral pathways. LTP was induced in a full glutamate receptor blockade (100 µM D-AP5, 10 µM NBQX, 500 µM R,S-MCPG, and 10 µM LY341495) by paired stimulation. Only one pathway (black box) was active during paired stimulation, the other pathway served as a control (white boxes). (**B**) Average fold change in the EPSP slope plotted against time for both the control and paired pathways. Sample EPSP traces shown are averages of 10 traces from the
*Figure 3 continued on next page*

*Figure 3 continued*

paired (solid line) and control (broken red line) pathway taken at four time points (1-4) from a single experiment (scale bar: 4.0 mV by 40 ms for paired pathway EPSP, 4. 9 mV by 40 ms for control pathway EPSP). Stimulation artifacts have been removed for clarity. (C) Average fold change in EPSP slope plotted for control experiments in which glutamate receptor antagonists were applied alone, in the absence of paired stimulation (drugs only group). Note that drug washout was incomplete. Group data and averages plotted for (D) fold changes in EPSP slope and (E) changes in PPR across experiments as measured 30 min following paired stimulation and drug washout. EPSP slope was higher in the paired pathway than in the control pathway, and was associated with a decrease in PPR, suggesting that LTP had been induced in the paired pathway, and had a presynaptic locus of expression. Error bars represent S.E.M ($n = 5$–7 cells per condition). Asterisks denotes significance differences between groups (**$p<0.01$; *$p<0.05$; Wilcoxon matched pairs signed rank test or Kruskal-Wallis with post-hoc Dunn's test). N.S. denotes no significant differences between groups.

DOI: https://doi.org/10.7554/eLife.29688.008

The following figure supplements are available for figure 3:

**Figure supplement 1.** Anti-causal pairing of pre- and postsynaptic activity fails to induce $LTP_{pre}$ in full glutamate receptor blockade.

DOI: https://doi.org/10.7554/eLife.29688.009

**Figure supplement 2.** Induction of $LTP_{pre}$ under full glutamate receptor blockade in acute hippocampal slices.

DOI: https://doi.org/10.7554/eLife.29688.010

**Figure supplement 3.** Membrane depolarization rescues LTP induction in NMDAR blockade.

DOI: https://doi.org/10.7554/eLife.29688.011

stimulation to drive postsynaptic spiking. This protocol produced robust potentiation of the recorded EPSP (*Figure 3—figure supplement 3A,D*), and an increase in presynaptic efficacy (*Figure 3—figure supplement 3E*). As in previous studies, NMDAR inhibition with AP5 abolished LTP induction, including its presynaptic component of expression (*Figure 3—figure supplement 3B,D*). However, we found that if we augmented the levels of postsynaptic depolarization by current injection during tetanic stimulation, then LTP induction in AP5 was rescued, at least presynaptically (*Figure 3—figure supplement 3C,E*). These findings suggest that the importance of postsynaptic NMDAR signalling in $LTP_{pre}$ induction is to provide a source of depolarization rather than any necessary source of synapse-specific signalling. These findings also underscore the importance of taking the level of postsynaptic depolarization into consideration when LTP is induced following the blockade of one or more glutamate receptor class.

Collectively, our findings suggest that the role of glutamate signalling (including postsynaptic NMDAR signalling) in $LTP_{pre}$ induction is to drive postsynaptic depolarization. Physiologically, this means that a presynaptic terminal could in principle be potentiated if it fails to release glutamate, provided that its activity coincides with postsynaptic spiking, which could be triggered by glutamate release at other co-active synapses.

## LTP induction in glutamate receptor blockade is associated with an increase in $P_r$

We then returned to $Ca^{2+}$ imaging to determine whether we could directly observe increases in $P_r$ at single synapses associated with the induction of $LTP_{pre}$ in full glutamate receptor blockade (*Figure 4*). Because spine $Ca^{2+}$ transients, in contrast to EPSPs, are resilient to partial AMPAR blockade (*Emptage et al., 2003*), we found that they recovered well following drug washout, despite the difficulties associated with washing out high concentrations of NBQX (*Holbro et al., 2010*). Consistent with electrophysiological findings, causal pairing of pre- and postsynaptic spiking in full glutamate receptor blockade produced robust and reliable increases in $P_r$ ($\Delta P_r$: $0.38 \pm 0.07$; $n = 8$ spines; vs 0: $p<0.01$; *Figure 4A–C*). No such changes were elicited by drug application in the absence of pairing ($\Delta P_r$: $0.01 \pm 0.02$; $n = 9$ spines; vs. causal pairing: $p<0.01$), or by either presynaptic stimulation alone ($\Delta P_r$: $-0.03 \pm 0.03$; $n = 8$ spines; vs. causal pairing: $p<0.001$), or postsynaptic stimulation alone ($\Delta P_r$: $-0.00 \pm 0.03$; $n = 8$ spines; vs. causal pairing: $p<0.001$), or when postsynaptic spiking preceded, rather than followed, presynaptic stimulation during pairing ($\Delta P_r$: $-0.02 \pm 0.05$; $n = 8$ spines; vs. causal pairing: $p<0.001$) (*Figure 4B,C*). The induction of $LTP_{pre}$ in the absence of glutamatergic signalling was therefore Hebbian, requiring presynaptic activity to be causally paired with postsynaptic spiking.

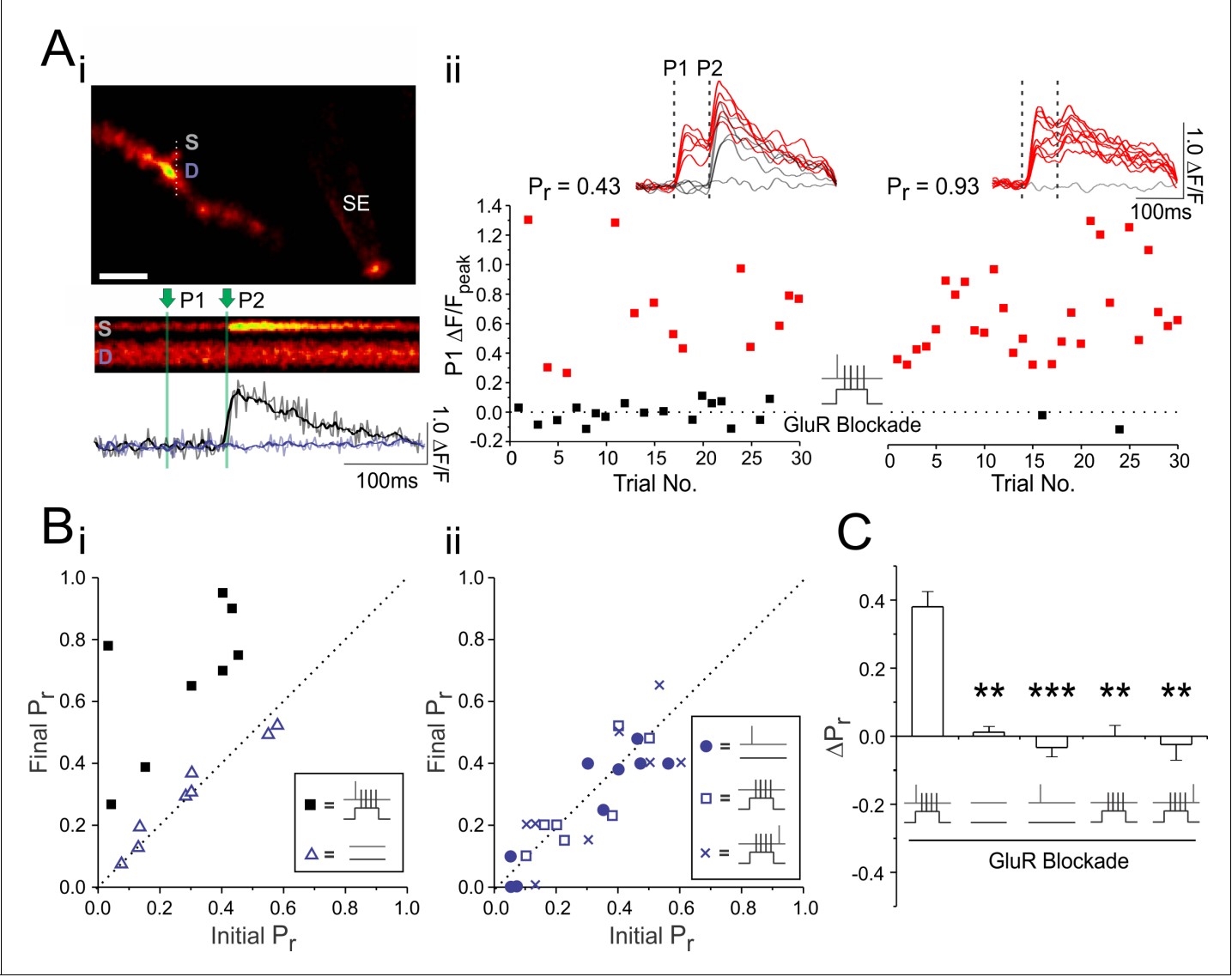

**Figure 4.** LTP induction in glutamate receptor blockade is associated with an increase in $P_r$. (**Ai**) Example experiment. Top, an image of a CA1 neuronal dendrite loaded with Oregon Green BAPTA-1 $Ca^{2+}$-sensitive dye with a stimulating electrode (SE) placed close to the dendrite in order to activate synapses within the vicinity (scale bar: 5 µm). Example of $Ca^{2+}$ transients were evoked by two stimulation pulses (P1 and P2; green vertical lines) are shown in the spine head (S) and underlying dendrite (D). Below the line scan are raw (grey) and smoothed (colored) traces quantifying fluorescence changes (ΔF/F) in the spine (black trace) and dendrite (purple trace). (**Aii**) The peak ΔF/Fs in the spine head following the first of the two stimulation pulses were plotted across 30 paired pulse stimulation trials given at baseline and 30 min following paired stimulation, which was delivered in full glutamate receptor blockade (50 µM D-AP5, 10 µM NBQX, 500 µM R,S-MCPG, and 10 µM LY341495). Red squares denote fluorescent increases above noise. Smoothed $Ca^{2+}$ traces from the last 10 trials are shown above each graph. $P_r$ was calculated as the proportion of total stimulation trials in which the first pulse resulted in a fluorescent increase above noise. (**B**) Group data. For each experiment, the imaged synapse's initial $P_r$ is plotted against its final $P_r$, calculated 30 min following one of five different stimulation paradigms. The broken diagonal line represents the expected trend if $P_r$ is unchanged across conditions. Only causal pairing of pre- and postsynaptic activity generated increases in $P_r$. (**C**) Average change in $P_r$. Error bars represent S.E.M. (n = 8–10 spines per condition). Asterisks denotes significance differences between the first group in the graph (**p<0.01; ***p<0.001; Kruskal-Wallis with post-hoc Dunn's test).

DOI: https://doi.org/10.7554/eLife.29688.012

## Postsynaptic depolarization promotes dendritic release of NO in a L-VGCC-dependent manner

We next investigated the mechanism by which paired stimulation could trigger increases in $P_r$ in the absence of glutamatergic signalling. The requirement for postsynaptic depolarization in the

induction of $LTP_{pre}$ suggests a need for a diffusible retrograde messenger. One promising, albeit still controversial, retrograde signal implicated in $LTP_{pre}$ induction is nitric oxide (NO) (for review see [*Padamsey and Emptage, 2014*]). Although NO synthesis has classically been associated with the activation of postsynaptic NMDARs (*Garthwaite and Boulton, 1995*), there is some suggestion that $Ca^{2+}$ influx from L-type voltage-gated $Ca^{2+}$ channels (L-VGCCs), which have previously been implicated in $LTP_{pre}$ (*Bayazitov et al., 2007*; *Zakharenko et al., 2001*), could trigger NO production (*Pigott and Garthwaite, 2016*; *Sattler et al., 1999*; *Stanika et al., 2012*); though definitive proof of a causal link between L-VGCC activation and NO synthesis at Schaffer-collateral synapses is lacking. We reasoned that if NO synthesis in CA1 neuronal dendrites can be triggered by L-VGCC activation, then NO production could occur in a manner dependent on postsynaptic depolarization, but independent of synapse-specific glutamatergic signalling.

To test this, we first asked whether $LTP_{pre}$, induced in glutamate receptor blockade, was dependent on L-VGCC activation and NO signalling. In keeping with our hypothesis, we found that pairing-induced increases in $P_r$ ($\Delta P_r$: 0.34 ± 0.04; n = 10 spines; p<0.01) were reliably abolished by bath application of the L-VGCC antagonist nitrendipine (20 µM) ($\Delta P_r$: −0.03 ± 0.04; n = 8 spines; vs. blockade: p<0.001) and by the NO scavenger carboxy-PTIO (cPTIO), either bath applied (50–100 µM) ($\Delta P_r$: −0.01 ± 0.04; n = 8 spines; vs. blockade: p<0.01) or injected into the postsynaptic neuron ($\Delta P_r$: −0.04 ± 0.06; n = 8 spines; vs. blockade: p<0.001) (*Figure 5A*). We confirmed our findings in acute slices, and found that nitrendipine and cPTIO blocked presynaptic enhancements induced under glutamate receptor blockade (*Figure 5—figure supplement 1*), suggesting that, as in cultured slices, presynaptic efficacy in acute slices was similarly regulated by L-VGCC and NO signalling.

We then examined whether NO production could be driven by postsynaptic depolarization in a L-VGCC-dependent manner. We transiently patched onto CA1 neurons in order to load them with the conventionally used NO-sensitive dye, DAF-FM (250 µM bolus-loaded), and then measured fluorescent changes in the apical dendrites prior to and following postsynaptic depolarization in glutamate receptor blockade. Given the poor signal-to-noise ratio associated with DAF-FM imaging, we drove strong postsynaptic depolarization by elevating extracellular $K^+$ to 45 mM, as previously described (*Sattler et al., 1999*; *Stanika et al., 2012*). Under these conditions, we observed increases in fluorescence in neuronal dendrites (*Figure 5B,C*). These increases were dependent on NO synthesis as they could be prevented by postsynaptic injection of cPTIO ($\Delta F/F$; control vs. cPTIO: 0.38 ± 0.04 vs. −0.03 ± 0.05; n = 5 cells/condition; p<0.05) or bath application of the NO synthase (NOS) inhibitor L-NAME ($\Delta F/F$: 0.00 ± 0.05; n = 5 cells; vs. control: p<0.05). Importantly, fluorescent increases were reliably abolished with nitrendipine ($\Delta F/F$: −0.02 ± 0.06; n = 5 cells; vs. control: p<0.05) (*Figure 5B,C*), suggesting that NO synthesis required L-VGCC activation.

We then attempted to image NO release in response to more physiologically-relevant forms of postsynaptic stimulation, such as the complex spikes we were using to induce LTP. To do so, we pre-loaded slices with the NO-sensitive dye 1,2-Diaminoanthraquinone (DAQ; 100 µg/mL), as previously described (*Chen et al., 2001*). We then patched onto a single cell and imaged the DAQ-associated changes in the cell after stimulating the cell with 600 complex spikes, delivered at 5 Hz (*Figure 5D*). Stimulation was performed in full glutamate receptor blockade. This protocol took advantage of the fact that DAQ forms an insoluble fluorescent precipitate upon reacting with NO, meaning that fluorescence would accumulate with stimulation and not readily wash away (*von Bohlen und Halbach et al., 2002*). We found that with 600 complex spikes, the accumulated NO signal in the dendritic arbour was sufficiently large to detect by our setup (*Figure 5D,E*). Notably, no signal was detected in the absence of any stimulation ($\Delta F/F$; stimulated vs. unstimulated: 2.97 ± 0.48; vs 0.22 ± 0.73; n = 9,7 cells; p<0.05), or when stimulation was delivered in the presence of nitrendipine ($\Delta F/F$: −0.14 ± 0.65; n = 7 cells; vs. control: p<0.05) or L-NAME ($\Delta F/F$: 0.24 ± 0.43; n = 8 cells; vs. control: p<0.05). These findings suggest that postsynaptic depolarization alone can drive NO release from neuronal dendrites in a L-VGCC dependent manner.

## NO release triggers increases in $P_r$ at active presynaptic terminals

Once NO is released, is it alone sufficient to induce $LTP_{pre}$ at active presynaptic terminals? To address this, we examined whether increases in $P_r$ could be elicited when presynaptic stimulation was paired with rapid photolytic release of NO (0.5–1 mM $RuNOCl_3$), in the absence of postsynaptic depolarization. We used $Ca^{2+}$ imaging to determine basal $P_r$ at a single synapse. We then paired

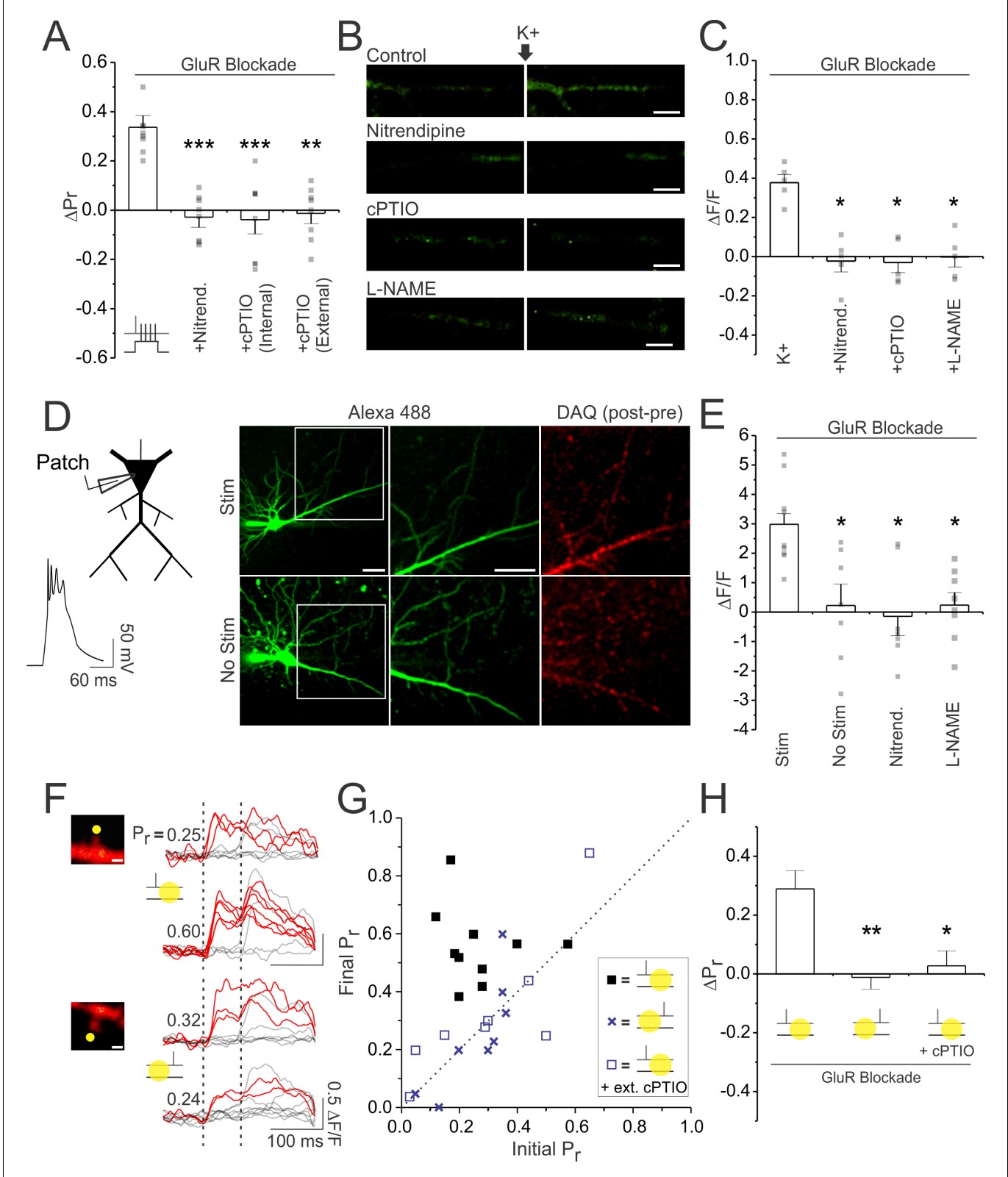

**Figure 5.** Postsynaptic depolarization increases $P_r$ by promoting dendritic release of NO in a manner dependent on L-VGCCs. (A) Average changes in $P_r$. Paired stimulation was delivered in full glutamate receptor blockade (50 μM D-AP5, 10 μM NBQX, 500 μM R,S-MCPG, and 10 μM LY341495), under control conditions, following treatment with the L-VGCC antagonist nitrendipine (20 μM), or following either the bath (100 μM) or intracellular (5 mM bolus-loaded) application of the NO scavenger cPTIO. (B) Images of CA1 apical dendrites loaded with NO-sensitive dye DAF-FM (250 μM bolus-
*Figure 5 continued on next page*

*Figure 5 continued*

loaded), prior to and following K$^+$ (45 mM) mediated depolarization (scale bar: 5 μm) in control conditions, or in the presence of nitrendipine, cPTIO, or the NO synthesis inhibitor L-NAME (100 μM; pre-incubation). K$^+$ stimulation evoked NO-sensitive and L-VGCC dependent increases in DAF-FM fluorescence. (C) Average K$^+$ induced fluorescence change (ΔF/F) of DAF-FM in apical dendrites. (D) Slices were preloaded with the NO-sensitive dye DAQ. Right. Example images of CA1 pyramidal neurons patched and loaded with Alexa Fluor 488 are shown, along with the associated change in DAQ fluorescence (post-pre stimulation images) recorded either following stimulation with 600 complex spikes (Stim) or following no stimulation (No stim). Images in white box are magnified in the adjacent image (scale bar: 5 μm). Left. Example of the complex spikes used during stimulation. (E) Average K$^+$ induced fluorescence change (ΔF/F) of DAF-FM in apical dendrites. Stimulation evoked NO-sensitive and L-VGCC dependent increases in DAQ fluorescence. (F) Samples of 10 superimposed Ca$^{2+}$ traces evoked in imaged spines (scale bar: 1 μm) by paired pulse stimulation (broken vertical bars); red traces depict successful release events to the first of the two pulses. Samples are taken from baseline and 25–30 min following a stimulation paradigm. The paradigm consisted of delivering presynaptic stimuli either (top) 7–10 ms before or (bottom) 7–10 ms after NO photolysis at the synapse (yellow circle); photolysis occurred in glutamate receptor blockade and in the absence of postsynaptic depolarization. P$_r$ was calculated as the proportion of total stimulation trials in which the first pulse resulted in a successful release event. In some experiments NO photolysis was conducted in cPTIO. Only causal pairings of presynaptic activity and NO release led to increases in P$_r$. (G) Final P$_r$ measured 25–30 min following the stimulation paradigm plotted against the initial P$_r$ for each synapse. The broken diagonal line represents the expected trend if P$_r$ is unchanged across conditions. (H) Average change in P$_r$. Error bars represent S.E.M. (n = 5–13 per condition). Asterisks denote significant differences from the control group (***p<0.001; **p<0.01; *p<0.05; Kruskal-Wallis with post-hoc Dunn's test).
DOI: https://doi.org/10.7554/eLife.29688.013

The following figure supplements are available for figure 5:

**Figure supplement 1.** Paired stimulation fails to induce LTP$_{pre}$ in acute hippocampal slices when L-VGCC or NO signalling is blocked.
DOI: https://doi.org/10.7554/eLife.29688.014

**Figure supplement 2.** Causal pairing of NO photolysis with presynaptic stimulation induces LTP$_{pre}$ of EPSPs recorded in acute slices.
DOI: https://doi.org/10.7554/eLife.29688.015

30–60 presynaptic stimuli, delivered at 5 Hz in full glutamate receptor blockade, with brief photolysis of NO, which was targeted to the spine head in order to emulate postsynaptic NO release. As with our standard LTP induction protocol, pairing was causal, with each NO photolysis event timed to occur 7–10 ms after each presynaptic stimulus. Under these conditions, we found significant increases in P$_r$ when assessed 30 min post-pairing (ΔP$_r$: 0.29 ± 0.07; n = 10 spines; p<0.01; *Figure 5F–H*). No such changes were produced when pairing occurred in the presence of bath-applied cPTIO (ΔP$_r$: 0.03 ± 0.05; n = 8 spines; vs. causal pairing: p<0.05; *Figure 5G,H*), suggesting that LTP$_{pre}$ did not result from non-specific effects associated with photolysis. Remarkably, when pairing was reversed such that presynaptic stimuli followed NO photolysis, no significant change in P$_r$ was observed (ΔP$_r$: −0.01 ± 0.04; n = 8 spines; vs. causal pairing: p<0.01; *Figure 5F–H*), suggesting that NO-mediated potentiation was Hebbian, requiring presynaptic activity to precede, rather than follow, NO release.

Previously, the effects of NO on synaptic efficacy have primarily been examined by recording EPSPs in acute slices (*Padamsey and Emptage, 2014*). We therefore sought to confirm our findings using NO photolysis in the same preparation (*Figure 5—figure supplement 2*). We loaded CA1 pyramidal neurons in acute slices with caged NO (100 μM RuNOCl$_3$) while recording EPSPs in the presence of AP5. Wide-field photolysis was triggered using a 1 ms flash from a UV lamp. Causal pairings of presynaptic activity with photolysis resulted in an enhancement of the EPSP and a decrease in PPR. These increases were absent when pairings were anti-causal, or when pairings occurred in the presence of cPTIO, which instead resulted in a modest depression of the EPSP. These findings confirm that NO can trigger LTP$_{pre}$ provided that its release precedes rather than follows presynaptic activity.

## Inhibition of presynaptic NMDARs prevents decreases in P$_r$ induced by glutamate release

Our findings suggest that a presynaptic terminal need not release glutamate in order to become potentiated, provided that its activity precedes strong postsynaptic depolarization. In fact, in our initial experiments we found that glutamate release, if anything, inhibited LTP$_{pre}$ and promoted LTD$_{pre}$ (*Figures 1* and *2*). These findings, however, were based on elevating glutamatergic release probability at the synapse either by using high-frequency presynaptic bursts or glutamate photolysis. We therefore sought to examine whether endogenous glutamate release also had a similar effect of inhibiting LTP$_{pre}$ and promoting LTD$_{pre}$. To investigate, we conducted single-spine Ca$^{2+}$ imaging

experiments in control conditions and under glutamate receptor blockade to examine how changes in $P_r$ were affected by glutamate signalling. Remarkably, we found that increases in $P_r$ produced in glutamate receptor blockade were significantly larger than under control conditions ($\Delta P_r$; blockade vs. control: $0.34 \pm 0.04$ vs. $0.18 \pm 0.02$; n = 10 spines; p<0.05; *Figure 6A–C*), suggesting that even endogenously released glutamate reduced elevations in $P_r$ induced by paired stimulation. We also examined the effects of glutamate receptor blockade on unpaired stimulation, during which single presynaptic stimuli (60 pulses at 5 Hz) were delivered in the absence of postsynaptic depolarization. As before (*Figure 2E*), this protocol reliably induced decreases in $P_r$ at synapses with high release probabilities ($P_r$ >0.5) under control conditions (*Figure 6D*). However, no such decreases were observed in glutamate receptor blockade ($\Delta P_r$ blockade vs. control: $0.00 \pm 0.03$ vs. $-0.21 \pm 0.05$; n = 10, 9 spines; p<0.05; *Figure 6D,E*). These findings suggest that endogenous glutamate release depresses $P_r$ regardless of the level of postsynaptic depolarization. Across conditions, $P_r$ changes were always more positive compared to controls when glutamate signalling was inhibited.

How might glutamate release drive decreases in $P_r$? We have previously reported functional and immunohistological evidence for the existence of presynaptic NMDARs at CA3-CA1 synapses (*McGuinness et al., 2010*); notably, these receptors act as reliable detectors for uniquantal glutamate release (*McGuinness et al., 2010*), and have been implicated in LTD$_{pre}$ (*Rodríguez-Moreno et al., 2013*; *Rodríguez-Moreno and Paulsen, 2008*; *Min and Nevian, 2012*; *Nevian and Sakmann, 2006*; *Andrade-Talavera et al., 2016*; *Sjöström et al., 2003*). We therefore examined whether glutamate was acting on these receptors to drive decreases in presynaptic efficacy. Given the difficulties associated with selectively blocking pre-, as opposed to post-, synaptic NMDARs, several groups have investigated the role of presynaptic NMDARs in plasticity by comparing the effects of bath application of AP5 or MK-801, which blocks both pre- and postsynaptic NMDARs, with that of intracellular MK-801 application, which selectively blocks postsynaptic NMDARs (*Nevian and Sakmann, 2006*; *Corlew et al., 2008*; *Corlew et al., 2007*; *Cormier and Kelly, 1996*). We sought to use a similar approach. However, because MK-801 does not readily washout, and since postsynaptic NMDARs greatly contribute to spine $Ca^{2+}$ influx (*Grunditz et al., 2008*; *Emptage et al., 1999*; *Holbro et al., 2010*), we first examined whether the permanent loss of postsynaptic NMDAR signalling affected our ability to measure $P_r$ using postsynaptic $Ca^{2+}$ imaging. We found that at about 50% of synapses, NMDAR blockade reduced, but did not entirely abolish synaptically-evoked $Ca^{2+}$ transients (*Figure 6—figure supplement 1A-C*). The residual $Ca^{2+}$ transients were mediated by activation of voltage-gated $Ca^{2+}$ channels (VGCCs) in response to AMPAR-mediated depolarization, and could be used to accurately measure $P_r$ (*Figure 6—figure supplement 1D,E*). Importantly, the average $P_r$ of these synapses did not significantly differ from that of synapses lacking a residual $Ca^{2+}$ transient in NMDAR blockade ($\Delta P_r$; AP5-sensitive vs. AP5-insensitive: $0.42 \pm 0.07$ vs. $0.47 \pm 0.11$; n = 8 spines/condition; p=0.67; *Figure 6—figure supplement 1B,C*). These findings suggest that, in NMDAR receptor blockade, VGCC-dependent spine-$Ca^{2+}$ influx can be used as a means of calculating $P_r$ at a sizeable and representative proportion of presynaptic terminals; nonetheless the use of VGCC-dependent $Ca^{2+}$ transients presents an inevitable selection bias in our study.

Using VGCC-dependent spine $Ca^{2+}$ transients, we found that when both pre- and postsynaptic NMDARs were blocked by bath application of either AP5 or MK-801, paired stimulation triggered increases in $P_r$ ($\Delta P_r$: $0.34 \pm 0.03$; n = 18 spines) that were not significantly different from those produced in full glutamate receptor blockade (p>0.99), but that were greater than increases in $P_r$ produced under control conditions (p<0.01) (*Figure 6A–C*). Bath blockade of NMDARs, like glutamate receptor blockade, also blocked decreases in $P_r$ produced by unpaired stimulation ($\Delta P_r$: $-0.02 \pm 0.02$; n = 17 spines; vs. control; p<0.05; vs. blockade: p>0.99; *Figure 6D,E*). Notably, bath application of MK-801 produced similar effects as AP5, suggesting that the effects of NMDARs on $P_r$ are associated with its ionotropic, rather than metabotropic effects (*Nabavi et al., 2013*) ($\Delta P_r$; bath MK-801 vs. AP5 paired stimulation: $0.33 \pm 0.04$ vs. $-0.32 \pm 0.03$; n = 8,9 spines; p=0.47; bath MK-801 vs. AP5 unpaired stimulation: $0.00 \pm 0.03$ vs. $-0.04 \pm 0.03$; n = 8,9 spines; p=0.27; *Figure 6B,C*)

To then specifically block postsynaptic NMDARs, we bolus-loaded cells intracellularly with MK-801 (see Materials and methods). Since MK-801 can have off-target effects on voltage-gated channels (*Jaffe et al., 1989*; *Kim et al., 2015*), we ensured our loading protocol reliably abolished NMDAR-mediated EPSPs and NMDAR-mediated spine $Ca^{2+}$ transients without impacting L-type voltage gated $Ca^{2+}$ currents (*Figure 6—figure supplement 2*). In contrast to bath application of AP5 or MK-801, we found that with postsynaptic application of MK-801, increases in $P_r$ produced by

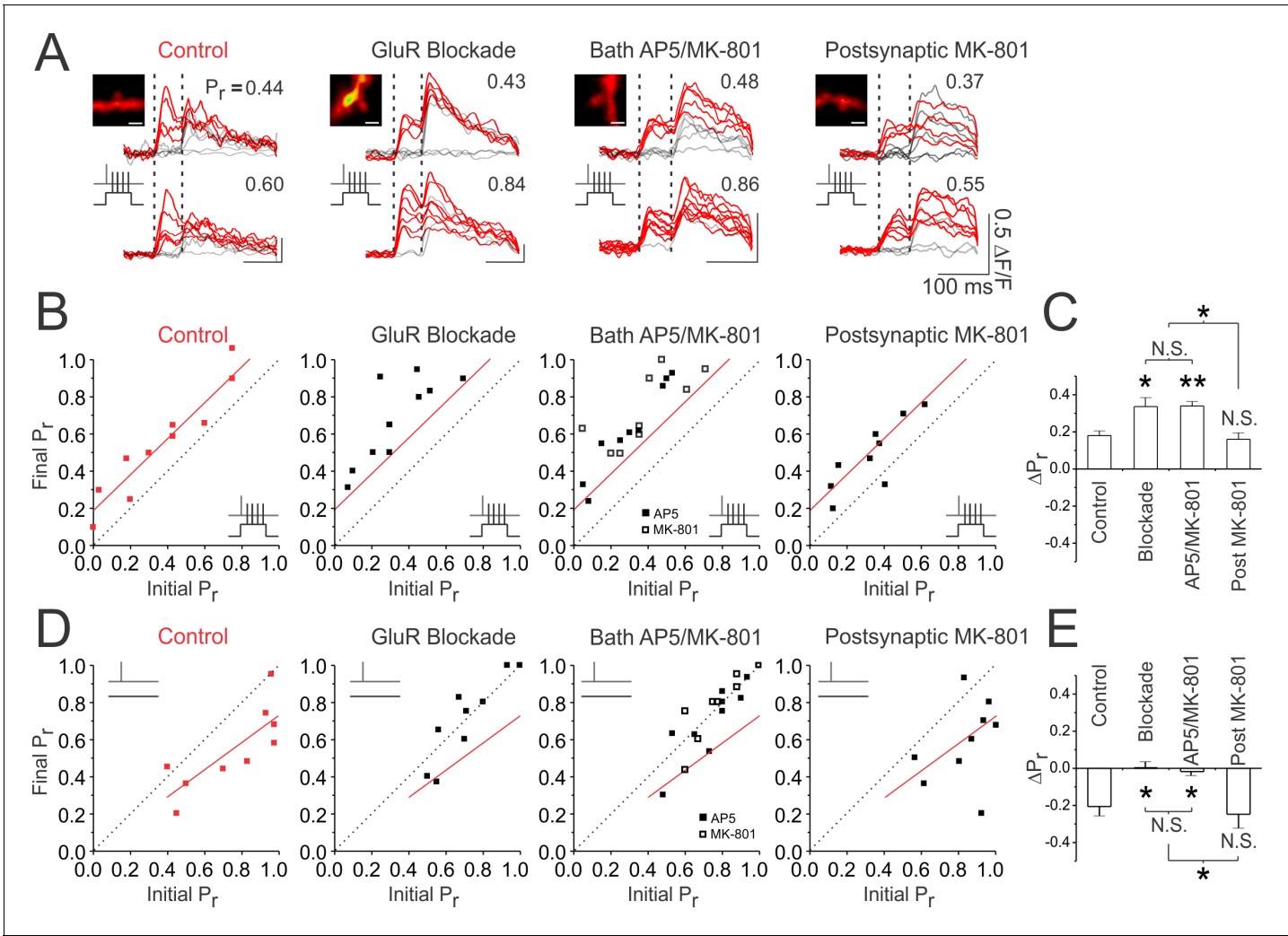

**Figure 6.** Inhibition of presynaptic NMDARs prevents decreases in $P_r$ induced by glutamate release. (A) Samples of 10 superimposed $Ca^{2+}$ traces evoked in imaged spines (white scale bar: 1 μm) by paired pulse stimulation (2 stimuli delivered 70 ms apart; represented by the vertical broken lines); red traces depict successful release events to the first of the two pulses. $P_r$ was calculated as the proportion of total stimulation trials in which the first pulse resulted in a successful release event. For each spine, sample $Ca^{2+}$ traces are shown during baseline and 25–30 min following paired stimulation. Experiments were also conducted with unpaired stimulation, in which presynaptic activity (60 pulses at 5 Hz) was delivered in the absence of postsynaptic depolarization. (B, D) Paired or unpaired stimulation was delivered in control conditions or following: full glutamate receptor blockade (50 μM D-AP5, 10 μM NBQX, 500 μM R,S-MCPG, and 10 μM LY341495), bath application of AP5 (50 μM) or MK-801 (20 μM), or intracellular application of MK-801 postsynaptically. For each experiment, the final $P_r$ measured 25–30 min following (B) paired or (E) unpaired stimulation is plotted against the initial $P_r$ measured at baseline. Red trendlines have been fitted to control data to guide visual comparison. The broken diagonal line represents the expected trend if $P_r$ is unchanged by stimulation. Inhibition of presynaptic NMDARs by full glutamate receptor blockade or by bath application of AP5 or MK-801 augmented increases in $P_r$ produced by paired stimulation and prevents decreases in $P_r$ produced by unpaired stimulation. (C,E) Average changes in $P_r$. Changes in $P_r$ were always more positive under conditions in which presynaptic NMDARs are blocked, suggesting that glutamate release drives decreases in $P_r$ via presynaptic NMDAR signalling. Error bars represent S.E.M. (n = 9–18 spines per condition). Asterisks denote significant differences (**$p<0.01$; *$p<0.05$; Kruskal-Wallis with post-hoc Dunn's test). N.S. denotes no significant difference. All comparisons were made against control data unless otherwise specified.

DOI: https://doi.org/10.7554/eLife.29688.016

The following figure supplements are available for figure 6:

**Figure supplement 1.** Spine $Ca^{2+}$ transients in NMDAR blockade are mediated by VGCCs, and can be used to accurately measure $P_r$.
DOI: https://doi.org/10.7554/eLife.29688.017

**Figure supplement 2.** Bolus loading of MK-801 intracellularly blocks NMDARs without affecting L-type voltage-gated $Ca^{2+}$ channels.
DOI: https://doi.org/10.7554/eLife.29688.018

**Figure supplement 3.** LTD$_{pre}$ does not require endocannabinoid signalling.
DOI: https://doi.org/10.7554/eLife.29688.019

*Figure 6 continued on next page*

*Figure 6 continued*

**Figure supplement 4.** Inhibition of presynaptic NMDARs augments LTP$_{pre}$ and abolishes LTD$_{pre}$.
DOI: https://doi.org/10.7554/eLife.29688.020

**Figure supplement 5.** Inhibition of presynaptic NMDARs augments LTP$_{pre}$ and abolishes LTD$_{pre}$ in acute slices.
DOI: https://doi.org/10.7554/eLife.29688.021

**Figure supplement 6.** Inhibition of presynaptic NMDARs rescues the effects of glutamate photolysis on presynaptic plasticity.
DOI: https://doi.org/10.7554/eLife.29688.022

paired stimulation ($\Delta P_r$: 0.16 ± 0.04; n = 9 spines; vs. control: p>0.99; *Figure 6A–C*) and decreases in $P_r$ produced by unpaired stimulation ($\Delta P_r$: −0.25 ± 0.07; n = 9 spines; vs. control: p>0.99; *Figure 6D,E*) did not significantly differ from control conditions (p>0.99), and were significantly different from changes in $P_r$ induced in glutamate receptor blockade (p<0.05) and extracellular NMDAR blockade (p<0.05). Collectively, these results suggest that glutamate release acts on pre-, but not post-, synaptic NMDARs to drive long-lasting decreases in $P_r$ observed during both paired and unpaired stimulation. Notably, these decreases were independent of endocannabinoid signalling (*Figure 6—figure supplement 3*).

We confirmed the effects of presynaptic NMDARs on presynaptic plasticity using electrophysiological recordings, both in organotypic (*Figure 6—figure supplement 4*) and acute slices (*Figure 6—figure supplement 5*). Using PPR changes to monitor presynaptic plasticity, we found that bath, but not intracellular blockade of NMDARs augmented LTP$_{pre}$ and abolished LTD$_{pre}$, consistent with our Ca$^{2+}$ imaging results.

Our findings that presynaptic NMDAR activation depresses $P_r$ would explain why glutamate photolysis in our earlier experiments inhibited LTP$_{pre}$ and promoted LTD$_{pre}$ (*Figure 2*). To confirm this, we repeated our photolysis experiments in the presence of MK-801, either intracellularly or extracellularly applied, again to differentially block pre- and postsynaptic NMDAR signalling (*Figure 6—figure supplement 4*). Consistent with our hypothesis, we found that bath, but not intracellular, application of MK-801 blocked the inhibitory effects of photolysis on LTP$_{pre}$ during paired stimulation, and prevented photolysis from inducing LTD$_{pre}$ during unpaired stimulation.

## Selective knockout of presynaptic NMDARs prevents decreases in $P_r$ induced by glutamate release

To directly assess the involvement of pre- and postsynaptic NMDARs in presynaptic plasticity, we differentially targeted these receptors for genetic deletion. We cultured hippocampal slices from a mouse line in which the gene encoding GluN1, the obligatory NMDAR subunit, was floxed (*Grin1$^{fx/fx}$*). We then virally injected Cre recombinase either into the CA3 or CA1 region to knockout pre- or postsynaptic NMDARs at Schaffer-collateral synapses. NMDAR currents were selectively abolished in targeted regions by 15 days post-injection (*Figure 7—figure supplement 1*).

We first examined plasticity in Cre-injected and control *Grin1$^{fx/fx}$* slices using electrophysiology (*Figure 7A*). In these experiments PPR was measured throughout the experiment. Paired stimulation resulted in LTP (fold $\Delta$EPSP$_{slope}$; control: 1.63 ± 0.08; CA3 KO: 2.09 ± 0.17; CA1 KO: 1.28 ± 0.06; n = 14, 12, 12 cells; p<0.001/condition; *Figure 7B*) with a presynaptic component of expression ($\Delta$PPR; control: −0.23 ± 0.04; CA3 KO: −0.47 ± 0.06; CA1 KO: −0.22 ± 0.02; n = 13, 12, 12 cells; p<0.001/condition; *Figure 7C*) that was evident across conditions, regardless of whether pre- or postsynaptic NMDARs were knocked out. PPR decreased by 5 min after plasticity induction (p<0.05; *Figure 7A*), and continued to decrease over the duration of the recording, in line with previous findings that the expression of LTP$_{pre}$ evolves over time (*Bayazitov et al., 2007*). Notably, slices lacking presynaptic NMDARs showed the greatest magnitude of LTP and the largest decrease in PPR, suggesting that LTP$_{pre}$ expression was strongest in this condition (fold $\Delta$EPSP$_{slope}$; CA3 KO vs. control: p<0.05; CA3 KO vs CA1 KO: p<0.01; *Figure 7B*) ($\Delta$PPR; CA3 KO vs. control: p<0.01; CA3 KO vs CA1 KO: p<0.01; *Figure 7C*). LTP magnitude of control slices exceeded that of slices lacking postsynaptic NMDARs (p<0.05), although PPR changes were of comparable magnitude in both conditions (p>0.99), suggesting that loss of postsynaptic NMDARs likely only impaired post-, but not pre-, synaptic plasticity. Collectively, these findings confirm that LTP$_{pre}$ induction is impaired by presynaptic NMDAR activation, and does not strictly require postsynaptic NMDAR activation.

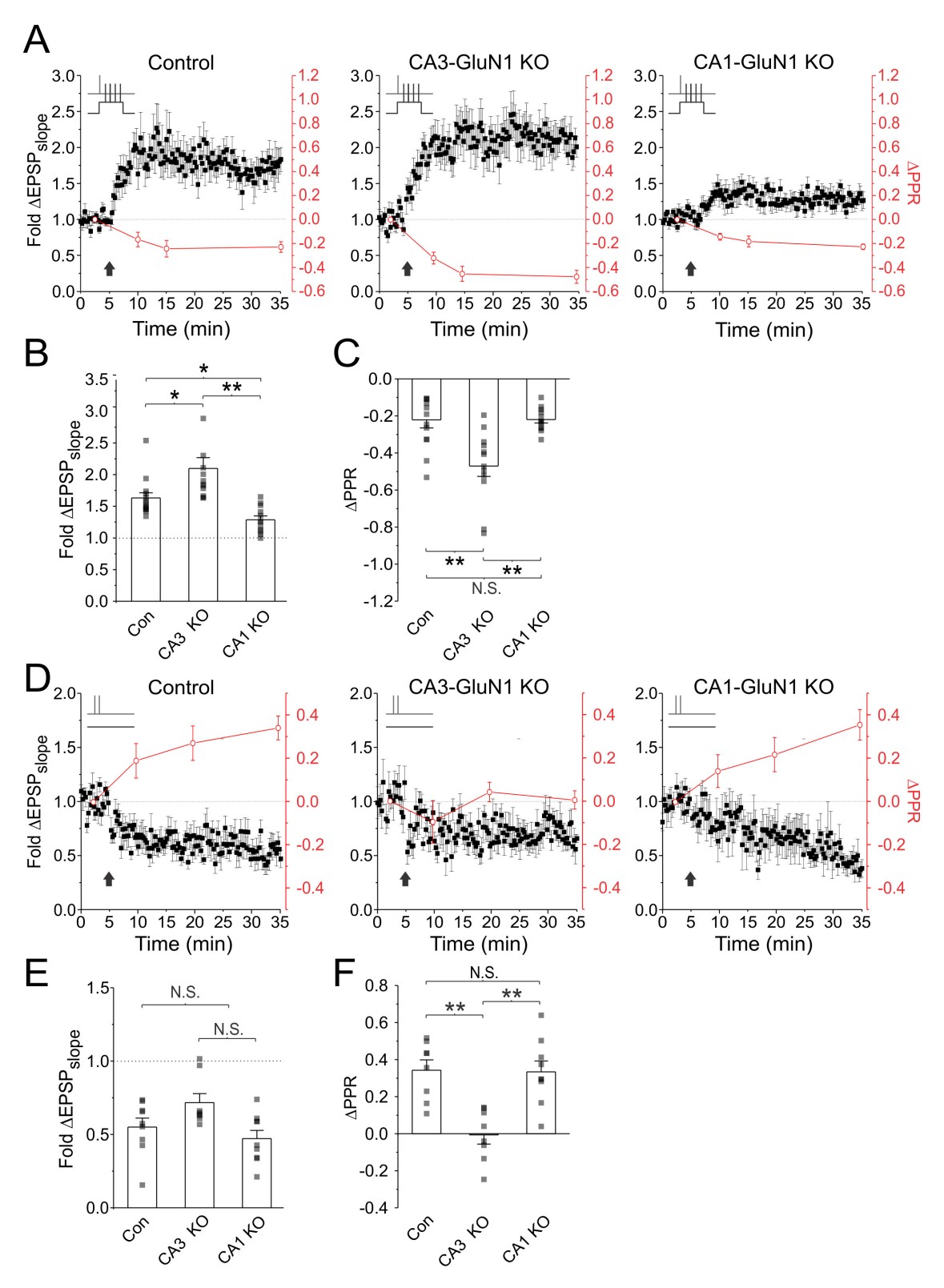

**Figure 7.** Genetic knockout of presynaptic NMDARs augments LTP_pre and abolishes LTD_pre. Cre recombinase was injected either into the CA3 or CA1 region of hippocampal slices cultured from floxed *Grin1* mice (*Grin1*$^{fx/fx}$) in order to selectively knockout pre- or postsynaptic NMDARs at Schaffer-collateral synapses. Slices were recorded electrophysiologically. (A,D) Average fold change in EPSP slopes (black) and PPR (red) plotted against time for plasticity induction across conditions. Plasticity was induced either using paired stimulation to induce LTP, or unpaired stimulation (2 pulses at 200 Hz,

*Figure 7 continued on next page*

*Figure 7 continued*

repeated 60 times at 5 Hz) to induce LTD. (**B,E**) Average EPSP slope changes across experiments 30 min post induction. (**C,F**) Average PPR changes across experiments 30 min post induction. Knockout of pre-, but not post-, synaptic NMDARs augmented decreases in PPR produced by paired stimulation, and abolished increases in PPR produced by unpaired stimulation alone. Knockout of presynaptic NMDAR signalling therefore augmented $LTP_{pre}$ and abolished $LTD_{pre}$. Error bars represent S.E.M. (n = 8–14 cells per condition). Asterisks denote significant differences from (\*\*p<0.01; \*p<0.05; Kruskal-Wallis with post-hoc Dunn's test). N.S. denotes no significant differences.

DOI: https://doi.org/10.7554/eLife.29688.023

The following figure supplement is available for figure 7:

**Figure supplement 1.** Selective knockout of pre- and postsynaptic NMDARs.

DOI: https://doi.org/10.7554/eLife.29688.024

We also examined LTD in *Grin1*$^{fx/fx}$ slices. To induce LTD, we used our unpaired, high-frequency burst stimulation protocol (2 pulses at 200 Hz repeated 60 times at 5 Hz; as in *Figure 1E*). This protocol produced robust depression of recorded EPSPs across conditions (fold $\Delta EPSP_{slope}$; control: 0.55 ± 0.06; CA3 KO: 0.65 ± 0.05; CA1 KO: 0.47 ± 0.06; n = 9, 9, 8 cells; p<0.01/condition; *Figure 7D*). However, increases in PPR were only seen in control ($\Delta PPR$ = 0.34 ± 0.05; n = 8) and postsynaptic NMDAR knockout slices ($\Delta PPR$ = 0.33 ± 0.05; n = 8; vs. control: p>0.99), and were absent in presynaptic NMDAR knockout slices ($\Delta PPR$ = 0.00 ± 0.05; n = 8; vs. control: p<0.01; vs. CA1 KO: p<0.01; *Figure 7D,F*). These findings confirm that pre-, but not post-, synaptic NMDARs are essential for $LTD_{pre}$ induction. Notably, loss of presynaptic NMDARs did not abolish LTD of the EPSP, suggesting that a postsynaptic component of LTD was likely still present in this condition. Changes in PPR, when present, were observed by 5 min following LTD induction (p<0.05; *Figure 7D*) and increased across the duration of the experiment, suggesting that the expression of $LTD_{pre}$, like that of $LTP_{pre}$, evolved over time.

Lastly, we used $Ca^{2+}$ imaging to directly examine changes in $P_r$ at single synapses in *Grin1*$^{fx/fx}$ slices (*Figure 8*). In these experiments we assessed $P_r$ at multiple time points following plasticity induction. Consistent with electrophysiological results, we found that genetic knockout of presynaptic NMDARs led to greater increases in $P_r$ following paired stimulation ($\Delta P_r$: CA3 KO vs. control: 0.37 ± 0.03 vs 0.20 ± 0.02; n = 12 spines/condition; p<0.001; *Figure 8A–C*), and abolished decreases in $P_r$ triggered by unpaired stimulation ($\Delta P_r$: CA3 KO vs. control: 0.00 ± 0.04 vs −0.45 ± 0.05; n = 12, 11 spines; p<0.0001; *Figure 8E–G*). These effects were evident within 5–15 min after plasticity induction (p<0.05), and were maintained throughout the 45 min post-induction imaging period (p<0.01; Figure D,H). We aligned changes in $P_r$ with time course measurements of PPR obtained in electrophysiological experiments (*Figure 7A,D*) and found good agreement between both measures following paired (*Figure 8D*) and unpaired (*Figure 8H*) stimulation. Collectively, these findings confirm that glutamate release drives decreases in $P_r$ via activation of presynaptic NMDARs. Such decreases occur independent of the level of postsynaptic depolarization; across conditions, changes in $P_r$ following paired and unpaired stimulation were always more positive when presynaptic NMDAR signalling was absent.

## Discussion

In this study we explored presynaptic plasticity at CA3-CA1 synapses. Based on our findings we present an entirely novel framework of presynaptic plasticity in which changes in $P_r$ at active presynaptic terminals are driven by two processes: (1) Hebbian activity, which promotes increases in $P_r$ via L-VGCC- and NO- dependent signalling, and (2) glutamate release, which promotes decreases in $P_r$ via presynaptic NMDAR activation. Both processes operate together to tune presynaptic function during synaptic activity, with net changes in $P_r$ depending on the strength of each process (*Figure 9*).

Consistent with this model, we found that when glutamate release was made reliable, either by presynaptic burst stimulation or by glutamate photolysis, Hebbian activity failed to drive net increases in $P_r$. Consequently, for presynaptic potentiation to occur, a presynaptic neuron must not only fire together with its postsynaptic partner, but it must also fail to release glutamate.

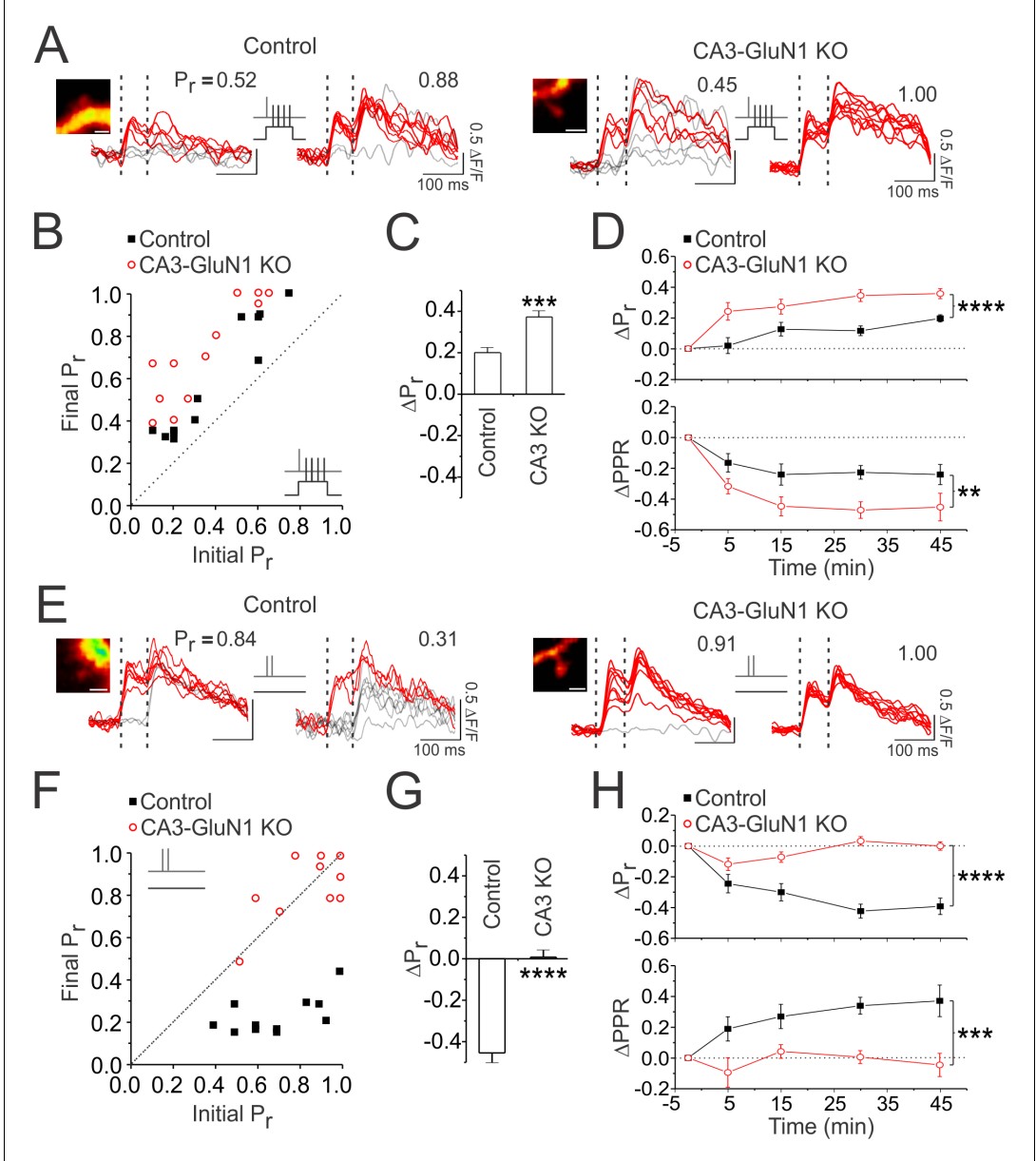

**Figure 8.** Selective knockout of presynaptic NMDARs prevents decreases in $P_r$ induced by glutamate release. (**A,E**) Samples of 10 superimposed $Ca^{2+}$ traces evoked in imaged spines (white scale bar: 1 μm) by paired pulse stimulation (2 stimuli delivered 70 ms apart; represented by the vertical broken lines); red traces depict successful release events to the first of the two pulses. $P_r$ was calculated as the proportion of total stimulation trials in which the first pulse resulted in a successful release event. For each spine, sample $Ca^{2+}$ traces are shown during baseline and 30–45 min following (**A**) paired or (**E**) unpaired stimulation (2 pulses at 200 Hz repeated 60 times at 5 Hz). (**B, F**) Group data. For each experiment, the final $P_r$ following (**B**) paired or (**F**) unpaired stimulation is plotted against the initial $P_r$ measured at baseline. The broken diagonal line represents the expected trend if $P_r$ is unchanged by stimulation. Knockout of presynaptic NMDARs augmented increases in $P_r$ produced by paired stimulation and prevented decreases in $P_r$ produced by unpaired stimulation. (**C,G**) Average changes in $P_r$ 30–45 min following (**C**) paired or (**G**) unpaired stimulation. (**D,H**) Average changes in $P_r$ at baseline, and 5, 15, 30, and 45 min following (**D**) paired and (**H**) unpaired stimulation. Changes in $P_r$ were time matched with changes in PPR recorded in electrophysiological experiments (*Figure 7*); both measures show good agreement of presynaptic changes. Changes in $P_r$ typically evolved over time, and were always more positive when presynaptic NMDARs were knocked out, suggesting that presynaptic NMDAR signalling was driving decreases in $P_r$. Error bars represent S.E.M. (n = 11–12 spines per condition). Asterisks denote significant differences (****p<0.0001; ***p<0.0001; **p<0.01; *p<0.05; Mann-Whitney test). All comparisons were made against control data unless otherwise specified.

DOI: https://doi.org/10.7554/eLife.29688.025

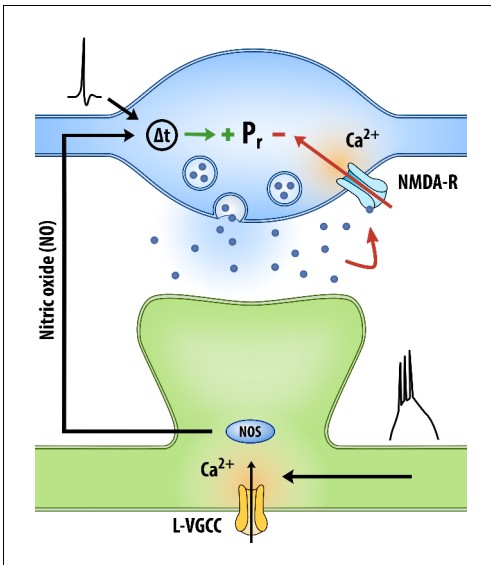

**Figure 9.** Proposed model of presynaptic plasticity. Changes in $P_r$ at active presynaptic terminals are determined by two opponent processes. (1) Increases in $P_r$ are driven by Hebbian activity, during which strong postsynaptic depolarization triggers the release of NO from neuronal dendrites. NO synthesis is dependent on NO synthase (NOS), activation of which is triggered by $Ca^{2+}$ influx through L-VGCCs. Importantly, NO drives an increase in $P_r$, but only at presynaptic terminals whose activity precedes its release. The detection of such an event requires an unidentified Hebbian detector ($\Delta t$) that is sensitive to the relative timings of NO release and presynaptic activity. (2) Decreases in $P_r$ are driven by glutamate release, via presynaptic NMDAR signalling. Net changes in $P_r$ depend on the strength of each process, and therefore on the amount of Hebbian signalling and glutamate release present during synaptic activity.
DOI: https://doi.org/10.7554/eLife.29688.026

## Experimental techniques

Our study is robust because we examine presynaptic plasticity under a diverse range of experimental conditions. We used both $Ca^{2+}$ imaging and PPR to assess presynaptic plasticity in cultured and acute hippocampal slices using a number of pharmacological and genetic manipulations. With such diverse experimental techniques, preparations, and manipulations, we found consistent support for the proposed model of presynaptic plasticity.

PPR and $Ca^{2+}$ imaging are markedly different techniques to assess presynaptic efficacy, each with its own assumptions, advantages, and disadvantages. $Ca^{2+}$ imaging provides a powerful means to monitor $P_r$ at single synapses in brain slices (*Padamsey and Emptage, 2011*). The excellent signal-to-noise with which this technique can be used to detect uniquantal glutamate release makes $P_r$ estimates robust to large changes in postsynaptic $Ca^{2+}$. Indeed, either removal of extracellular $Mg^{2+}$ (*Figure 2—figure supplement 1*) or bath application of AP5 (*Figure 2—figure supplement 1*), which more than doubles or halves the $Ca^{2+}$ transient respectively, has no effect on $P_r$ calculations made at resting membrane potential. Nonetheless, $Ca^{2+}$ imaging may bias synapse selection, particularly in favour of synapses producing large $Ca^{2+}$ transients (*Padamsey and Emptage, 2011*). Such selection bias, conversely, is absent in PPR calculations, which reflect aggregate changes in $P_r$ over a larger number of stimulated synapses. However, PPR is an indirect measures of $P_r$, and may be confounded by factors such as postsynaptic receptor desensitization (*Yang and Calakos, 2013*); though, not at the interpulse intervals used in this study (*Arai and Lynch, 1998*). Despite these caveats, in our study, results from PPR measurements and $Ca^{2+}$ imaging were consistent with one another across a range of experimental conditions, making it unlikely that our assessment of presynaptic plasticity was confounded. Both of these techniques, therefore, present valid means of measuring presynaptic efficacy, at least in the context of this study.

## LTP$_{pre}$ can occur in the absence of glutamate receptor signalling

The proposed model provides a mechanism by which presynaptic terminals releasing little or no glutamate can become potentiated provided that their activity is accompanied by strong postsynaptic depolarization. Notably, most central synapses have low glutamate release probabilities, with some synapses appearing to release no glutamate in response to presynaptic stimulation (*Voronin and Cherubini, 2004*; *Stevens, 2003*). This is true for synapses recorded in both *in vitro* and *ex vivo* preparations from young and adult rodents. In fact, under electron microscopy, a significant portion of synapses (up to 35–50%) in the adult rodent hippocampus have presynaptic zones lacking synaptic vesicles in their near proximity (<170 nm); these so-called 'nascent zones' have been hypothesized to be functionally silent (*Bell et al., 2014*). Although the existence of *bona fide* presynaptically silent synapses remains controversial (*Voronin and Cherubini, 2004*), the low release probabilities

(average $P_r$ of approximately 0.2 [*Murthy et al., 1997*]) of central synapses suggests that it is possible that activity at a presynaptic terminal may not elicit glutamate release at the synapse, but may still coincide with strong postsynaptic depolarization, driven by glutamate release at other co-active synapses. Under such conditions, the mechanisms proposed in this study could enable presynaptic induction of Hebbian potentiation at these synapses.

Our finding that presynaptic enhancements can occur without glutamatergic signalling at the synapse raises the question as to why many studies show that LTP induction can be abolished or impaired by blockade of one or more glutamate receptor subtypes (*Holbro et al., 2010*; *Collingridge et al., 1983*; *Bashir et al., 1993*). To address this question, it is first necessary to recognize that not all LTP induction protocols are associated with presynaptic enhancements (*Padamsey and Emptage, 2014*). This is because $LTP_{pre}$ induction requires higher levels of postsynaptic depolarization than $LTP_{post}$ induction (*Padamsey and Emptage, 2014*; *Zakharenko et al., 2003*; *Bayazitov et al., 2007*). Whether presynaptic enhancements are obtained will therefore depend on the levels of postsynaptic depolarization achieved during LTP induction, which in turn will be influenced by a variety of experimental factors, including the frequency and intensity of stimulation (*Padamsey and Emptage, 2014*). Nonetheless, even studies reporting $LTP_{pre}$ also find that inhibition of glutamate receptors, in particular NMDARs, abolish or reduce presynaptic enhancements (*Ryan et al., 1996*; *Ratnayaka et al., 2012*; *Emptage et al., 2003*; *Bliss and Collingridge, 2013*; *Enoki et al., 2009*; *Nikonenko et al., 2003*; *Stanton et al., 2005*; *Padamsey and Emptage, 2014*; *Zakharenko et al., 2003*; *Bayazitov et al., 2007*; *Zakharenko et al., 2001*). In such cases it is important to recognize that AMPARs, KARs, NMDARs, and mGluRs can all contribute to postsynaptic depolarization (*Grienberger et al., 2014*; *Schiller and Schiller, 2001*; *Grover and Yan, 1999b*; *Chemin et al., 2003*). Given that presynaptic changes rely on the voltage-dependent release of NO, it is possible that blockade of any of these glutamate receptor classes would abolish or reduce $LTP_{pre}$ in an indirect way, by reducing postsynaptic depolarization and the activation of L-VGCCs. This may explain, in part, why experimental manipulations that augment the levels of postsynaptic depolarization reliably rescue LTP in AMPAR (*Holbro et al., 2010*; *Fuenzalida et al., 2010*), NMDAR (*Padamsey and Emptage, 2014*; *Grover and Teyler, 1992*; *Zakharenko et al., 2003*; *Bayazitov et al., 2007*; *Zakharenko et al., 2001*; *Kullmann et al., 1992*; *Huber et al., 1995*; *Grover et al., 2009*; *Morgan and Teyler, 2001*), and mGluR blockade (*Wilsch et al., 1998*). Critically, our LTP induction protocol used strong postsynaptic depolarization, which was elicited by somatic current injection, and therefore independent of synaptic activity. This circumvented the need for any glutamate receptor-dependent depolarization during paired stimulation and enabled us to directly assess the function of glutamate signalling in $LTP_{pre}$, independent of its effects on postsynaptic depolarization. Based on these results, we argue that the physiological role of glutamate release in $LTP_{pre}$ is for driving postsynaptic spiking as opposed to conveying a synapse-specific signal; this contrasts with the role of glutamate release in postsynaptic plasticity, in which synapse-specific activation of postsynaptic NMDARs is necessary for $LTP_{post}$ induction.

While our approach for inducing LTP resembles that of traditional STDP protocols, which rely on NMDAR activation (*Dan and Poo, 2004*), there are two key differences. Firstly, in our study, postsynaptic depolarization took the form of complex spikes, which included a brief period (7–10 ms) depolarization before the first spike (see Materials and methods). This period of subthreshold depolarization is known to facilitate the induction of LTP, possibly by inactivating voltage-gated $K^+$ channels within the dendrite, which otherwise impede action potential backpropagation (*Watanabe et al., 2002*; *Gasparini et al., 2007*; *Hoffman et al., 1997*; *Johnston et al., 1999*; *Migliore et al., 1999*; *Sjöström and Häusser, 2006*). Secondly, like complex spikes recorded *in vivo* (*Ranck, 1973*), the spike trains we triggered contained broadened action potentials, which likely reflect strong depolarization in the dendrites (*Hoffman et al., 1997*; *Migliore et al., 1999*). Consequently, the postsynaptic waveforms used in our study were likely to generate greater levels of postsynaptic depolarization, and in a manner independent of glutamate release and NMDAR activation, than those used in traditional STDP studies.

## $LTP_{pre}$ requires nitric oxide signalling

It has long been recognized that the induction of $LTP_{pre}$ requires a retrograde signal (*Williams et al., 1989*). One promising candidate is NO (*Garthwaite and Boulton, 1995*). The role of NO in plasticity has been a source of much controversy, and some studies have concluded that

NO signalling is not necessary in LTP induction (for review see [*Padamsey and Emptage, 2014*]). However, given that NO is likely to be important for presynaptic strengthening, the effect of NO signalling on synaptic plasticity will depend on whether presynaptic enhancements are obtained following LTP induction (*Padamsey and Emptage, 2014*). Indeed, studies that actually confirm presynaptic changes following LTP induction, including our own, consistently demonstrate that presynaptic enhancements depend on the synthesis and release of NO in both acute and cultured hippocampal preparations (*Ratnayaka et al., 2012*; *Nikonenko et al., 2003*; *Stanton et al., 2005*; *Johnstone and Raymond, 2011*).

It has generally been assumed that NO synthesis is dependent on $Ca^{2+}$ influx from postsynaptic NMDARs (*Garthwaite and Boulton, 1995*); however, several studies, including our own, have demonstrated that induction of $LTP_{pre}$ is possible in NMDAR blockade, suggesting that a NMDAR-dependent NO signalling pathway is not required for $LTP_{pre}$ (*Zakharenko et al., 2003*; *Bayazitov et al., 2007*; *Zakharenko et al., 2001*). Here, we provide direct evidence for an alternative pathway for NO synthesis that is crucial for presynaptic strengthening, and that is driven by strong postsynaptic depolarization via the activation of L-VGCCs. Why L-VGCC-, as opposed to NMDAR-, mediated NO signalling is specifically required for $LTP_{pre}$ is not known, but may result from differences in the magnitude, kinetics, and/or spatial extent of NO signalling associated with L-VGCC and NMDAR activation. Unfortunately, the poor sensitivity of NO-indicator dyes makes this possibility difficult to currently investigate.

It has previously been shown that exogenous NO can potentiate synaptic transmission, and that this potentiation is restricted to synapses that are active during NO release (*Arancio et al., 1996*; *Zhuo et al., 1993*). Here, we extend these findings by showing that photolysis of NO at single synapses can directly drive increases in Pr, and that this increase can occur in the absence of glutamatergic signalling. Moreover, we demonstrate that the potentiating effects of NO are not only restricted to active synapses, but specifically at synapses whose activity precede, rather than follow, NO release; thus, the requirements of NO signalling are consistent with those of Hebbian and spike-timing dependent plasticity (*Dan and Poo, 2004*). These findings also suggest the existence of a Hebbian detector at the presynaptic terminal that is sensitive to the timing between presynaptic activity and NO release; at least one isoform of guanylate cyclase is sensitive to NO in a $Ca^{2+}$-dependent manner, making it a potential candidate for integrating NO signalling and presynaptic activity (*Zabel et al., 2002*).

Although our study focussed on phasic NO signalling, LTP may additionally require a tonic, low-level of NO signalling (*Hopper and Garthwaite, 2006*). It will be important to examine the differential roles of tonic and phasic NO signaling in presynaptic plasticity in future studies. Moreovoer, while we provide evidence in support of NO as a retrograde signal in $LTP_{pre}$, it may not be the only retrograde signal involved. Indeed, neurotrophic factors, transsynaptic signals, as well as contact-dependent processes are all known to regulate $P_r$ (*Regehr et al., 2009*); whether such signals play a role in $LTP_{pre}$ induction remains to be elucidated.

## Glutamate drives $LTD_{pre}$ by activating presynaptic NMDARs

At active presynaptic terminals, whereas Hebbian activity drives increases in $P_r$, we show, unexpectedly, that glutamate release drives decreases in $P_r$ by acting on presynaptic NMDARs. Using both pharmacological and genetic manipulations, we found that presynaptic NMDAR signalling operated both during $LTP_{pre}$ and $LTD_{pre}$ induction paradigms to reduce $P_r$. Our finding suggests that the potentiating effects of Hebbian activity and the depressing effects of endogenous glutamate release occur concurrently during synaptic activity. Thus, the processes underlying $LTP_{pre}$ and $LTD_{pre}$ induction do not act independently as originally believed, but operate jointly to tune synaptic function. Our results may explain why sometimes the same pairing protocol that produces $LTP_{pre}$ at low $P_r$ synapses, produces $LTD_{pre}$ at high $P_r$ synapses; presumably the level of Hebbian activity achieved by such protocols is not of sufficient magnitude to prevent the depressing effects of glutamate release at high $P_r$ synapses (*Hardingham et al., 2007*; *Sáez and Friedlander, 2009*). Our results may also explain why the locus of LTP expression, whether pre- or postsynaptic, appears to depend on initial $P_r$ (*Larkman et al., 1992*). With higher basal release probabilities, more glutamate is released for a given LTP induction protocol, meaning that $LTP_{post}$ is favoured owing to greater postsynaptic NMDAR-signalling, whereas $LTP_{pre}$ is inhibited owing to greater presynaptic NMDAR-signalling.

Thus, low $P_r$ synapses will have a tendency to express LTP presynaptically, while high $P_r$ synapses will have a tendency to express LTP postsynaptically (*Larkman et al., 1992*).

In contrast to our findings, inhibition of presynaptic NMDARs at neocortical synapses does not appear to effect LTP magnitude (*Rodríguez-Moreno et al., 2013*; *Rodríguez-Moreno and Paulsen, 2008*; *Sjöström et al., 2003*). It is possible that the low frequency (0.2 Hz) of presynaptic stimulation used during LTP induction in these studies does not elicit sufficient glutamate release to drive decreases in $P_r$ via presynaptic NMDAR activation.

Studies using STDP protocols, however, have found a role for presynaptic NMDARs in the induction of $LTD_{pre}$ at neocortical synapses (*Min and Nevian, 2012*; *Nevian and Sakmann, 2006*; *Sjöström et al., 2003*; *Sjöström et al., 2007*). This form of $LTD_{pre}$ is thought to additionally require endocannabinoid receptor 1 (CB1R) signalling. Although we also found presynaptic NMDARs to be necessary for $LTD_{pre}$ induction, we found no requirement for CB1Rs. However, the protocol we used to induce $LTD_{pre}$ was not a STDP protocol, and did not involve postsynaptic spiking, which is thought to be necessary to drive endocannabinoid release (*Min and Nevian, 2012*). Instead, our protocol used presynaptic stimulation, either in the form of single or short bursts of action potentials, delivered in the absence of postsynaptic depolarization. *Rodríguez-Moreno et al., 2013* similarly found that patterned presynaptic stimulation delivered in the absence of postsynaptic spiking induced $LTD_{pre}$ at neocortical synapses, and in a manner independent of CB1R activity. Such findings suggest that under some experimental conditions, glutamate release from presynaptic terminals alone is sufficient to induce $LTD_{pre}$ by acting on presynaptic NMDARs without the additional need for endocannabinoid signalling.

NMDARs can have both metabotropic and ionotropic receptor signalling capacities (*Dore et al., 2016*). We found that blocking ionotropic signalling with bath, but not postsynaptic, application of MK-801 was sufficient to prevent glutamate from driving decreases in $P_r$. Combined with our conditional NMDAR knockout experiments, these findings suggest that ionotropic presynaptic NMDAR signalling is necessary for depressing $P_r$. Blockade of presynaptic NMDAR function with MK-801 similarly abolishes $LTD_{pre}$ in neocortex induced by STDP protocols (*Rodríguez-Moreno et al., 2013*; *Rodríguez-Moreno and Paulsen, 2008*; *Rodríguez-Moreno et al., 2011*). A recent paper by (*Carter and Jahr, 2016*), however, failed to find functional evidence for presynaptic NMDARs in the neocortex (but see [*Abrahamsson et al., 2017*]), and showed that instead, metabotropic signalling by postsynaptic NMDARs was responsible for spike-timing dependent LTD (*Carter and Jahr, 2016*). The locus of LTD expression, however, was not assessed in this study. Given that metabotropic receptor signalling from postsynaptic NMDARs is believed to underlie the induction of $LTD_{post}$ (*Nabavi et al., 2013*), and based on our current findings, we would hypothesize that ionotropic presynaptic NMDAR signalling will preferentially play a role in $LTP_{pre}$ and $LTD_{pre}$ induction.

Previously we have demonstrated that presynaptic NMDARs at hippocampal synapses facilitate transmitter release during theta stimulation (*McGuinness et al., 2010*). When considered with our current findings, presynaptic NMDARs appear to be important for presynaptic facilitation in the short-term, but presynaptic depression in the long-term. This is consistent with the finding that presynaptic NMDARs in the neocortex similarly mediate short-term plasticity of glutamate release, and yet are similarly implicated in $LTD_{pre}$ (*Min and Nevian, 2012*; *Sjöström et al., 2003*; *Corlew et al., 2008*). It may appear peculiar for a single protein to mediate seemingly disparate functions; however, another way to view the presynaptic NMDAR is as a dynamic regulator of presynaptic activity, appropriately tuning glutamate release depending on the patterns of pre- and postsynaptic activity. As such, the receptor may aid glutamate release during theta-related activity, but, triggers $LTD_{pre}$ when this release fails to elicit sufficiently strong levels of postsynaptic depolarization.

## Implications of the proposed model of presynaptic plasticity

In this study we present evidence for a novel model of presynaptic plasticity, in which changes in $P_r$ at presynaptic terminals depend on the levels of 1) Hebbian signalling and 2) glutamate release that accompany presynaptic activity (*Figure 9*). Critically, the levels of glutamate release at a synapse will not only depend on basal $P_r$, but also on the pattern of presynaptic activity and on the state of the synapse (e.g. facilitating or depressing) (*Dobrunz et al., 1997*; *Dobrunz and Stevens, 1997*), which dictates how $P_r$ changes throughout a train of stimulation.

One interpretation of the proposed model is that presynaptic plasticity, by adjusting $P_r$, corrects any mismatch between two variables: 1) the likelihood that presynaptic activity is accompanied by

strong postsynaptic depolarization (i.e. Hebbian activity) and 2) the likelihood that presynaptic activity is accompanied by glutamate release. Accordingly, as we have shown in this study, increases in $P_r$ will preferentially occur when Hebbian activity is present at the synapse, but glutamate release is absent; whereas decreases in $P_r$ will preferentially occur when Hebbian activity is absent, but glutamate release is present. These scenarios reflect the correction of an otherwise profound mismatch that exists between the ability for presynaptic activity at a synapse to *drive* postsynaptic activity (reflected by the amount of glutamate release), and the ability for presynaptic activity to *predict* postsynaptic spiking (reflected by the amount of Hebbian signalling). We would hypothesize that $P_r$ would continue to change until these mismatches are corrected. This could explain why, for a given plasticity induction protocol, $P_r$ tends to a common equilibrium value across synapses (*Hardingham et al., 2007*); this value presumably reflects the point at which the levels of glutamate and Hebbian signalling associated with the stimulation protocol are equally matched.

A key implication of our model is that the pattern of presynaptic activity will substantially impact changes in $P_r$. As demonstrated in our study, when high frequency bursts of presynaptic stimulation are paired with postsynaptic spiking, $P_r$ remains low, whereas when single presynaptic stimuli are instead paired with postsynaptic spiking, $P_r$ potentiates to higher values (*Figure 1*). At high $P_r$ synapses, it is known that glutamate release is preferentially driven by single spikes, and otherwise depresses in response to high frequency bursting (*Dobrunz et al., 1997*; *Dobrunz and Stevens, 1997*). By contrast, at low $P_r$ synapses, glutamate release is preferentially driven by high frequency bursts, and is minimally responsive to single presynaptic spikes (*Dobrunz et al., 1997*; *Dobrunz and Stevens, 1997*). Thus, presynaptic plasticity appears to adjust $P_r$ such that glutamate release is preferentially driven by the pattern of presynaptic stimulation (bursts or singe spikes) that best predicts strong postsynaptic depolarization; $P_r$ is set low in the case of presynaptic bursts and high in the case of single presynaptic spikes. This is particularly relevant given that different patterns and frequencies of presynaptic firing are likely to convey different information (*Butts and Goldman, 2006*). Consequently, presynaptic plasticity would enable the presynaptic terminal to act as a dynamic filter by preferentially tuning $P_r$ to ensure that only information relevant for postsynaptic spiking is transmitted. Such a process would greatly enhance the signal-to-noise ratio of synaptic transmission.

## Materials and methods

All animal work was carried out in accordance with the Animals (Scientific Procedures) Act, 1986 (UK).

### Cultured hippocampal slices

Unless otherwise stated in the text, cultured hippocampal slices were used for imaging and electrophysiological experiments owing to the excellent optical and electrophysiological access to cells and synapses afforded by this preparation. Cultured hippocampal slices (350 μm) were prepared from male Wistar rats (P7-P8), as previously described (*Emptage et al., 2003*). Slices were maintained in media at 37°C and 5% $CO_2$ for 7–14 days prior to use. Media comprised of 50% Minimum Essential Media, 25% heat-inactivated horse serum, 23% Earl's Balanced Salt Solution, and 2% B-27 (ThermoFisher Scientific - Invitrogen, UK) with added glucose (6.5 g/L), and was replaced every 2–3 days. During experimentation, slices were perfused with artificial cerebrospinal fluid (ACSF; 1–2 mL/min), which was constantly bubbled with carbogen (95% $O_2$ and 5% $CO_2$) and heated to achieve near-physiological temperatures in the bath (31-33°C). ACSF contained (in mM) 145 NaCl, 16 $NaHCO_3$, 11 glucose, 2.5 KCl, 2–3 $CaCl_2$, 1–2 $MgCl_2$, 1.2 $NaH_2PO_4$, and, to minimize photodynamic damage, 0.2 ascorbic acid and 1 Trolox.

### Acute hippocampal slices

Acute hippocampal slices were used to confirm key findings in cultured hippocampal slices. When this preparation was used, it is clearly stated in the text and figure captions. Coronal acute hippocampal slices (400 μm) were prepared from 2 to 3 week old male Wistar rats. Tissue was dissected in a sucrose-based ACSF solution (in mM: 85 NaCl, 65 sucrose, 26 $NaHCO_3$, 10 glucose, 7 $MgCl_2$, 2.5 KCl, 1.2 $NaH_2PO_4$, and 0.5 $CaCl_2$). The whole brain was sliced into coronal sections using a Microm HM 650V vibratome (Thermo Scientific, UK). Hippocampal tissue were allowed to recover at room

temperature in normal ACSF (120 NaCl, 2.5 KCl, 2 CaCl$_2$, 1 MgCl$_2$, 1.2 NaH$_2$PO$_4$, 26 NaHCO$_3$, and 11 glucose), which was bubbled with 95% O$_2$ and 5% CO$_2$. Slices were given at least 1 hr to recover before use. During experimentation, slices were perfused with ACSF (3 mL/min) containing picrotoxin (100 µM; Sigma, UK). The ACSF was constantly bubbled with carbogen (95% O$_2$ and 5% CO$_2$) and heated to achieve near-physiological temperatures in the bath (31-33°C).

## GluN1 knockout

The GluN1 NMDAR obligatory subunit was selectively knocked out of CA3 or CA1 neurons to respectively remove either pre- or postsynaptic NMDAR function at the Schaffer-collateral synapses. Hippocampal slices were cultured from Grin1$^{fx/fx}$ mouse pups (P6-P8) (B6.129S4-Grin1$^{tm2Stl}$/J; Stock no. 005246; Jackson Laboratory, Bar Harbor, Maine, USA) in which both copies of the GluN1 encoding genes are floxed. After 1–2 days in culture, Cre recombinase (AAV1.hSyn.Cre.WPRE.hGH; Penn Vector) and a floxed variant of tdTomato (AAV1.CAG.Flex.tdTomato.WPRE.bGH; Allen Institute, Seattle, Washington, US) were co-injected into either the CA3 or CA1 region using a sharp glass pipette (100–120 MΩ) with its tip broken, coupled to a picospritzer (Science Products, Germany). For dense transfection of the CA3 region, a total of 75–150 nL of virus was injected over three sites at a high titer (Cre – 6.6 × 10$^{12}$ GC/mL; tdTomato – 2.94 × 10$^{12}$ GC/mL). For sparse transfection of CA1 cells, a single CA1 site was injected with 50 nL of virus at a lower titre. For controls, injections into CA3 or CA1 lacked Cre recombinase. Knockout was assessed by examining patch recordings of NMDAR currents at +40 mV in the presence of NBQX (10 µM; Abcam, UK) and picrotoxin (100 µM). NMDAR currents were abolished by 15 days post injection. Blind patch recordings in CA3 revealed that 91% (21/23) of cells lacked NMDAR currents, suggesting that injections had successfully infected the vast majority of cells in this region.

## Electrophysiological recordings

CA1 pyramidal neurons were recorded from either using low (4–8 MΩ) or high resistance patch electrodes (18–25 MΩ) filled with standard internal solution (in mM: 135 KGluconate, 10 KCl, 10 HEPES, 2 MgCl$_2$, 2 Na$_2$ATP and 0.4 Na$_3$GTP; pH = 7.2–7.4), or sharp microelectrodes (80–120 MΩ) filled with 400 mM KGluconate. In some experiments (*Figure 3—figure supplement 2*; *Figure 5—figure supplement 1*) low resistance (4–8 MΩ) patch electrodes were used containing an ATP regenerating internal solution in order to minimize the effects of postsynaptic dialysis (in mM: 130 KGluconate, 10 KCl, 10 HEPES, 10 NaPhosphocreatine, 4 MgATP, 0.4 Na$_3$GTP and 50 U/mL creatine phosphokinase; pH = 7.2–7.4) (*Kullmann et al., 1992*). The recording method used in a given experiment is indicated in the main text or the figure caption.

## Stimulation protocols

A glass electrode (4–8 MΩ), filled with ACSF, was placed in stratum radiatum. Continuous basal stimulation (0.05–0.10 Hz) was present for all experiments, and was only interrupted to deliver paired-pulse or tetanic stimulation. Stimulation intensity was adjusted to evoke a 5–10 mV EPSP; pulse duration was set at 100 µs. Paired-pulse stimulation, unless otherwise stated, consisted of 2 presynaptic stimuli delivered 70 ms apart.

Baseline recordings were kept short (approximately 5 min) when recording using low resistance patch electrodes (4–8 MΩ) to minimize the effects of dialysis. We found that LTP$_{pre}$ induction was impaired with longer baseline recordings. Indeed, we could induce LTP$_{pre}$ under NMDAR blockade following a 5 min baseline recording (*Figure 7A–C*) but not a 10 min baseline recording (5 vs. 10 min: fold ΔEPSP$_{slope}$: 1.91 ± 0.13 vs 0.87 ± 0.08; ΔPPR: −0.48 ± 0.08 vs −0.03 ± 0.03; n = 12 vs. 5 cells; p<0.01).

LTP induction consisted of 60 single pulses delivered at 5 Hz each paired with postsynaptic depolarization. Postsynaptic depolarization took the form of a complex spike. To emulate a complex spike, we injected a postsynaptic current waveform (2–3 nA) that was approximately 60 ms in duration and resulted in 3–6 spikes at ~100 Hz, with the first spike occurring 7–10 ms after the presynaptic stimulus. This was done by injecting a current waveform (2–3 nA) with a 7–10 ms rising phase, a 20 ms plateau phase, and a 30–33 ms falling phase. However, in experiments shown in *Figure 2*, the current waveform took the form of a 50 ms flat current step; although successful, this protocol led to poorer control of the start of postsynaptic bursting. LTP induction in both instances was more robust

when the cell was depolarized by approximately 10–15 mV from resting membrane potential (approx. −65 mV) during the 12 s induction protocol; this facilitated broad spiking during postsynaptic current injeciton. Stimulating electrodes were placed within 50–70 μm of the soma to ensure that postsynaptic depolarization reached stimulated synapses without significant attenuation. In glutamate receptor blockade experiments, the two stimulating electrodes used were placed at the same depth in the slice to ensure that drug washout rates were comparable in both pathways. In these experiments, if a strong monosynaptic IPSP was present following application of full glutamate receptor blockade, the experiment was omitted. For two pathway experiments, to ensure each electrode was stimulating independent populations of axons we used the collision test (*Lipski, 1981*). Briefly, each pathway was successively stimulated, 1–2 ms apart. If both axonal populations are perfectly overlapping, then successive stimulation should generate a synaptic response comparable to the stimulation of either pathway alone, owing to the axonal refractory period. If however, both axonal populations are perfectly independent, successive stimulation should generate an EPSP response comparable to the sum total of the EPSP generated by stimulation of either pathway alone.

LTD induction consisted of either 60 single or paired (inter-stimulus interval of 5 ms) presynaptic pulses delivered at 5 Hz in the absence of postsynaptic depolarization; single pulses only induced $LTD_{pre}$ at high $P_r$ synapses ($P_r \geq 0.5$; *Figures 2E* and *6D*), whereas paired pulses induced LTD at all synapses (*Figure 8F*). During either stimulation regime, the membrane was hyperpolarized (<-100 mV) to prevent somatic and dendritic spiking; stimulation intensity was also kept low to avoid spiking, such that basal EPSP amplitude did not typically exceed 5 mV.

## Electrophysiology and analysis

All electrophysiological data was recorded using WinWCP (Strathclyde Electrophysiology Software) and analyzed using Clampfit (Axon Insturments) and Excel (Microsoft). The initial EPSP slope, calculated during the first 2–3 ms of the response, was used to analyze changes in the EPSP throughout the recording. This was done to ensure only the monosynaptic component of the EPSP was analyzed. This is particularly important in cultured slices in which polysynaptic activity may confound EPSP amplitude measures. All data was normalized to the average EPSP slope recorded during baseline to yield ΔEPSP slope. Paired pulse ratio (PPR) was calculated as the average EPSP slope evoked by the second stimulation pulse divided by the average EPSP slope evoked by the first stimulation pulse, as previously described (*Kim and Alger, 2001*); averages were calculated from 5 to 10 paired pulse trials. Decreases in PPR are thought to reflect increases in release probability (*Schulz et al., 1994*).

## Confocal imaging

Confocal images were taken using a BioRad MRC-1000 confocal laser scanning system, controlled by LaserSharp software. A 488 nm argon laser line was used for fluorophore excitation. Images were acquired on an upright Olympus BX50WI microscope equipped with a 60x water-immersion objective (Olympus; 0.9 NA).

## Bolus loading

Bolus loading was used to fill CA1 neurons with dye or drugs whilst minimizing the amount of time the cell was patched on to. Loading was achieved by transiently patching onto cells (60 s) using low-resistance patch electrodes (4–8 MΩ) containing a high-concentration of drug or dye (see relevant sections of Materials and methods for exact concentrations) dissolved in standard internal solution. Slow withdrawal of the patch using a piezoelectric drive ensured re-sealing with no observable adverse effects to cell health. Cells were then subsequently re-patched for the purposes of delivering postsynaptic depolarization if and when required.

## Ca$^{2+}$ imaging and analysis

For Ca$^{2+}$ imaging, cells were bolus-loaded with OGB-1 (0.5–1 mM for 60 s) to enable $P_r$ measurements to be conducted in the absence of electrophysiological recordings, and associated dialysis. Cells were re-patched during LTP or LTD induction. A stimulating glass electrode (4–8 MΩ) was then brought near (5–20 μm) to a branch of imaged dendrite within stratum radiatum. For visualization purposes, electrode tip was coated with bovine serum albumin Alexa Fluor 488 conjugate

(ThermoFisher Scientific, Invitrogen, UK), as previously described (*Ishikawa et al., 2010*). Briefly, a 0.05% BSA-Alexa 488 solution was made with 0.1M phosphate-buffered saline containing 3 mM $NaN_3$. Pipette tips were placed in the solution for 2–5 min.

To find a synapse responsive to axonal stimulation, axons were stimulated with pairs of stimuli (2 pulses 70 ms apart) to increase the chances of eliciting a $Ca^{2+}$ response. During stimulation, laser scanning was initially restricted to a single line through a number of synapses on the dendrite to enable for rapid assessment of potentially responsive spines. Because stimulation intensity was kept low to prevent dendritic and somatic spiking, generally only one or two spines could be clearly identified as responding to stimulation; though only one spine was typically taken for experimentation since laser scanning had to be restricted to a line crossing both the spine and a region of underlying dendrite in order to determine if spine $Ca^{2+}$ signals were contaminated by dendritic or somatic spikes. Responsive synapses were always found in the vicinity of the stimulating electrode, which was placed within 100 μm of the soma. Synapse selection, however, was invariably biased in favour of mushroom spines, with head diameters ranging from 0.3 to 1.0 μm, as these synapses were clearly visible and produced larger $Ca^{2+}$ transients.

$Ca^{2+}$ images were acquired in line scan mode at a rate of 500 Hz and analyzed using ImageJ and Microsoft Excel. Increases in spine fluorescence ($\Delta F/F = F_{transient}–F_{baseline}/F_{baseline}$) following the delivery of the first stimulus is thought to reflect successful glutamate release from the presynaptic terminal (*Emptage et al., 2003*; *Emptage et al., 1999*). The proportion of successful fluorescent responses to the first stimulus across stimulation trials was used to calculate $P_r$. $P_r$ was assessed on the basis of 15–40 trials at baseline and at 25–30 min post-tetanus. For high $P_r$ synapses (>0.8) the number of stimulation trials was limited to 15–20 to avoid photodynamic damage that results from imaging the frequent $Ca^{2+}$ responses generated at these synapses. For all other synapses, $P_r$ was generally assessed using 20–35 trials of stimulation. Stimulation was kept of a sufficiently low intensity to avoid somatic and dendritic spiking. When spikes did occur, as evidenced by a simultaneous $Ca^{2+}$ rise in both the spine and the dendrite, a successful release event would require spine fluorescence to precede that of the dendrite, or to be of greater magnitude (*Nevian and Helmchen, 2007*). Synapses with initial $P_r$ values of 0–0.7 were used for LTP experiments, and synapses with $P_r$ values of 0.4–1.0 were used for LTD experiments. In experiments involving glutamate receptor blockade, $P_r$ was measured prior to drug application at baseline, and measured post-tetanus, following drug washout. In experiments involving NMDAR blockade, using either AP5 or MK-801, drugs were present for the duration of the experiment and, therefore, present for both the baseline and post-tetanus measurements of $P_r$. Experiments were excluded if the synapse became non-responsive and there was evidence of either substantial drift of the stimulation electrode or photodynamic damage (i.e. blebbing of the dendrite or sudden increases in basal fluorescence intensity).

## Nitric oxide imaging

Experiments involving DAF-FM (ThermoFisher Scientific, Invitrogen, UK) imaging were carried out in Tyrodes buffer (in mM: 120 NaCl, 2.5 KCl, 30 glucose, 4 $CaCl_2$, 0 $MgCl_2$, and 25 HEPES) containing 50 μM D-AP5, 10 μM NBQX, 500 μM MCPG, and 100 μM LY341495 (Abcam, UK) to block glutamate receptors, as well as 1 μM Bay K-8644 (Abcam, UK) to prevent L-VGCC desensitization during $K^+$ application as previously described (*Sattler et al., 1999*; *Stanika et al., 2012*). CA1 pyramidal neurons were bolus-loaded with 250 μM of DAF-FM for 60 s. Apical dendrites, often secondary or tertiary branches, within 100 μm of the soma were imaged at one focal plane, once prior to, and once 5–10 s following, the addition of a high $K^+$ Tyrodes solution (in mM: 32.5 NaCl, 90 KCl, 30 glucose, 4 $CaCl_2$, 0 $MgCl_2$, 25 HEPES, which included: 50 μM D-AP5, 10 μM NBQX, 500 μM MCPG, and 100 μM LY341495). Laser power and exposure was kept to a minimum to avoid photobleaching. In our hands, DAF-FM basal fluorescence was not quenched by intracellular addition of cPTIO.

1,2-Diaminoanthraquinone (DAQ; Sigma, UK) was used to image activity-dependent NO release under more physiological conditions. DAQ was loaded as previously described (*Chen et al., 2001*). DAQ was prepared as a 5 mg/mL stock solution dissolved in DMSO. Hippocampal slices cultures were treated with 100 μg/mL of the solution for 2 hr at 37°C and 5% $CO_2$. Slices were then placed on the rig, and perfused with heated (31-33°C) and carbogenated (95% $O_2$ and 5% $CO_2$) ACSF for 30 min prior to imaging to wash-off excess dye. DAQ was imaged in full glutamate receptor blockade using 488 nm excitation light and a 570 nm long-pass emission filter prior to and following stimulation of a single patched CA1 neuron with 600 complex spikes at 5 Hz (see Stimulation protocols

section). Control cells were left unstimulated. Following DAQ imaging, cells were re-patched, loaded with Alexa Fluor 488 (100 µM; ThermoFisher Scientific, Invitrogen, UK), and imaged. Alexa Fluor 488 fluorescence was used to determine the proportion of imaged DAQ fluorescence that co-localized to the recorded cell. DAQ fluorescence was compared before and after stimulation in the imaged cell.

## Photolysis

A 405 nm laser (Photonics, UK) was used for spot photolysis. The laser was focussed to a small spot (~1.2 µm diameter) by overfilling the back aperture of a 60x water-immersion lens (Olympus, UK). Electrode manipulators and recording chambers were mounted on a movable stage, which enabled a region above the spine head to be positioned beneath the photolysis spot. Laser exposure was controlled using a fast shutter (LS6; Uniblitz). For glutamate photolysis, MNI glutamate (Tocris, UK) was focally delivered through a glass pipette (4–8 MΩ; 10 mM MNI glutamate) using a picospritzer (Science Products, Germany). Laser exposure was limited to ~2 ms and, in each experiment, the laser intensity (0.5–2 mW) was adjusted to generate a $Ca^{2+}$ response in the underlying spine that was comparable to the response generated by electrical stimulation. Ruthenium nitrosyl chloride (RuN-OCl$_3$), which has sub-millisecond release kinetics (*Bettache et al., 1996*), was used for NO photolysis experiments. For spot photolysis, 0.5–1 mM RuNOCl$_3$ (Sigma) was bath applied and uncaged using 30–60 laser pulses (25 ms; 2 mW) delivered at 5 Hz; presynaptic stimulation either preceded or followed NO photolysis by 7–10 ms. Using the NO-indicator, DAF-FM (Invitrogen), we calibrated laser power to liberate approximately 10 nM of NO per pulse. We did this by targeting the soma of DAF-FM loaded neurons (250 µM bolus-loaded) for photolysis at different laser powers while recording the resulting increases in fluorescence using the confocal laser in line scan mode (500 Hz). We aimed for an increase in fluorescence of about 6–7% (averaged across several trials), which based on the manufacturer's data on the concentration-dependent fluorescence of DAF-FM, amounts to a release of approximately 10 nM of NO. For wide-field UV photolysis, 100 µM RuNOCl$_3$ was added to the patch electrode and allowed to diffuse into the cell for 10–15 min prior to commencing the experiment. A UV Flash Lamp (HI-TECH Scientific) was used to deliver a 1 ms wide-field uncaging pulse (100 V) that was timed to occur 7–10 ms before or after presynaptic stimulation. Because of the time required for the UV lamp to recharge between flashes, about 20 of the 60 presynaptic pulses delivered at 5 Hz were not associated with a flash.

## NMDAR pharmacology

In experiments requiring both pre- and postsynaptic NMDARs to be blocked, either D-AP5 (50–100 µM; Abcam, UK) or MK-801 (20 µM; Abcam, UK) was added in bath for the duration of the experiment. In the case of MK-801, slices were pre-incubated with the drug for at least 1 hr prior to experimentation. Experiments in which NMDARs were blocked with bath application of AP5 and with bath application of MK-801 produced similar results and so conditions were combined for data analysis. Postsynaptic NMDARs were blocked by bolus loading of 5 mM MK-801 for 60 s, after which 20 min was given for the drug to diffuse and take effect. This was the case for both imaging (*Figure 6*) and electrophysiology experiments (*Figure 7*). In the latter, cells were re-patched 20 min after bolus loading with normal internal solution; re-patching did not result in a notable intracellular washout of MK-801, likely reflecting the high affinity of drug binding (approximately 37 nM) (*Wong et al., 1986*). Most electrophysiological experiments in the literature use 1 mM of MK-801 in the patch electrode to block postsynaptic NMDARs (*Rodríguez-Moreno et al., 2013*; *Rodríguez-Moreno and Paulsen, 2008*; *Nevian and Sakmann, 2006*; *Rodríguez-Moreno et al., 2011*). We found that using this protocol, we failed to induce LTP using paired stimulation ($\Delta$EPSP$_{slope}$: 0.83 ± 0.13; vs. 1.0: p=0.22; $\Delta$PPR: 0.13 ± 0.08 vs. 0; p=0.19). However, this protocol may have resulted in more intracellular loading of MK-801 than our rapid (60 s) bolus loading approach, and may have therefore resulted in off-target effects. At high concentrations (>100 µM), MK-801 can inhibit voltage-gated $K^+$ and $Ca^{2+}$ channels (*Jaffe et al., 1989*; *Kim et al., 2015*). Indeed, we found that 1 mM patch loading of MK-801 for 5 min reduced L-VGCC-mediated $Ca^{2+}$ influx (isolated by using 10 µM mibefradil, 0.3 µM SNX-482, and 1 µM ɯ-conotoxin-MVIIC, as previously described [*Bloodgood and Sabatini, 2007*]) by approximately 50% ($\Delta$F/F: control vs. 1 mM MK-801: 1.15 ± 0.05 vs. 0.53 ± 0.06; p<0.01; *Figure 6—figure supplement 2D-F*). Our bolus loading procedure (5 mM MK-801 for 60 s), by

contrast, produced no change in L-VGCC function (ΔF/F: 1.10 ± 0.07; vs. control: p=0.99; vs. 1 mM MK-801: p<0.05; *Figure 6—figure supplement 2D-F*) despite effectively inhibiting NMDAR function (*Figure 6—figure supplement 2A-C*), and also failed to abolish the induction of LTP$_{pre}$ using our pairing protocol ( *Figure 6—figure supplement 4A-C, Figure 6—figure supplement 5A-C* ).

### Other pharmacology

Glutamate receptor blockade was achieved using D-AP5 (50–100 µM; Abcam, UK), NBQX (10 µM; Abcam, UK), R,S-MCPG (500 µM; Abcam, UK) and LY341495 (100 µM; Abcam, UK). L-VGCCs were blocked with nitrendipine (20 µM; Abcam, UK). NO synthase was inhibited by pre-incubation of slices with L-NAME (100 µM; Sigma, UK), which started at least 20 min prior to experimentation. Extracellular NO was scavenged by bath application of cPTIO (50–100 µM; Sigma, UK). Intracellular NO was scavenged by bolus loading cells with 5 mM cPTIO. Endocannabinoid signalling (CB1 receptor) was inhibited by bath application of the AM-251 (2 µM; Tocris, UK).

### Statistical analysis

Tests used to assess statistical significance are stated at the end of all Figure captions. Only non-parametric tests were used owing to small sample sizes (*Siegel, 1956*). For single comparisons, two-tailed Mann-Whitney or Wilcoxon matched pairs signed rank tests were used, depending on whether the data was unpaired or paired, respectively. Wilcoxon signed rank tests were also used to determine if data significantly differed from an expected value. For multiple comparisons, Kruskal-Wallis tests were used with post-hoc Dunn's tests. Means and standard error of the mean (S.E.M.) are represented in the text as mean ±S.E.M.

Sample sizes that were typical for the field (n = 5–15; independent experiments/biological replicates) were used in this study, and provided sufficient power (>80%) to detect the expected experimental effects reported in our study. For spine imaging, typically only one spine was imaged per cell per experiment. Samples were randomly assigned to conditions that were being concurrently run. Masking was not used for sample allocation or data collection. A single data point reflected the average of multiple measurements (technical replicates) within an experiment; this is detailed in the relevant Materials and methods sections.

## Acknowledgements

ZP was funded by a Clarendon Scholarship, a scholarship from the National Science and Engineering Research Council of Canada, and a Junior Research Fellowship from Magdalen College, University of Oxford. Experimental work was funded by grants from the Medical Research Council (MRC, UK) and the Biotechnology and Biological Sciences Research Council (BBSRC, UK). The funders had no role in study design, data collection and analysis, decision to publish, or preparation of the manuscript

## Additional information

### Funding

| Funder | Author |
| --- | --- |
| Medical Research Council | Nigel Emptage |
| Biotechnology and Biological Sciences Research Council | Nigel Emptage |

The funders had no role in study design, data collection and interpretation, or the decision to submit the work for publication.

### Author contributions

Zahid Padamsey, Conceptualization, Formal analysis, Supervision, Investigation, Visualization, Methodology, Writing—original draft, Writing—review and editing; Rudi Tong, Investigation, Formal analysis; Nigel Emptage, Data Curation, Resources, Supervision, Funding acquisition, Methodology, Writing—review and editing

**Author ORCIDs**
Zahid Padamsey (iD) https://orcid.org/0000-0001-9177-8210
Nigel Emptage (iD) http://orcid.org/0000-0002-7348-497X

**Decision letter and Author response**
Decision letter https://doi.org/10.7554/eLife.29688.029
Author response https://doi.org/10.7554/eLife.29688.030

## Additional files

**Supplementary files**
• Transparent reporting form
DOI: https://doi.org/10.7554/eLife.29688.027

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
