## [Decision Letter]

Thank you for submitting your article "Glutamate is required for depression but not potentiation of long-term presynaptic function" for consideration by *eLife*. Your article has been reviewed by two peer reviewers, and the evaluation has been overseen by a Reviewing Editor and Richard Aldrich as the Senior Editor. The following individuals involved in review of your submission have agreed to reveal their identity: Stanislav Zakharenko (Reviewer #1).

The reviewers have discussed the reviews with one another and the Reviewing Editor has drafted this decision to help you prepare a revised submission.

Summary:

A comprehensive understanding of Hebbian plasticity requires solving many open questions, including those addressed by Padamsey and colleagues. In their new study, the authors use an impressive variety of techniques to further investigate the mechanisms of long-term plasticity at hippocampal CA3-CA1 synapses at the level of a single synapse. The manuscript delivers two major conclusions. First, the presynaptic component of LTP (LTP_pre_) is induced by postsynaptic depolarization that is mediated by L-type voltage-gated calcium channels. Postsynaptic depolarization generates nitric oxide, a retrograde messenger that acts on a cognate presynaptic terminal to increase the probability of release (hence LTP_pre_). The second conclusion is that glutamate released by presynaptic terminals acts through presynaptic NMDARs and decreases the probability of release, without a contribution of postsynaptic NMDARs. The authors argue that LTP_pre_ depends on the basal level of probability of release.

Despite the importance of all topics touched in this work and their impressive number, the authors need to strengthen the *direct* evidence of the LTP_pre_ in the observed phenomena. Electrophysiological tools do not allow for unequivocal proof of LTP_pre_ / LTD_pre_ expression and modulation. Thus, the reviewers expect that the authors strengthen their data using optical tools to monitor CaT or synaptic vesicle exocytosis to support the major conclusions of the paper.

Please consider the detailed concerns below and write back with your plans to address these experimentally and an estimate of the time it will take to do so. We will share your response with the Board and reviewers who will then confer to issue their thoughts on your proposed experiments.

Essential revisions:

1) Why didn't the authors base the entire experimental design on the optical methodology they have developed in the past (Emptage et al., 2003; Emptage, Bliss and Fine, 1999)? Imaging the number of successful postsynaptic calcium transients (assuming that the responses to single quanta of glutamate are indeed reliably detected) would focus on presynaptic changes thus being much more direct and rewarding. This approach was used here but just as a confirmation of the electrophysiological results (Figure 2, Figure 4, Figure 5, Figure 6). Unfortunately, results from recordings of EPSPs and from optical experiments are not superimposable. Intracellular recordings of synaptic potentials sample large populations of synapses, a likely mix of different presynaptic and postsynaptic phenotypes and forms of plasticity. On the contrary, postsynaptic calcium transients sample a small subset of these, presumably biasing the selection toward spiny synapses located on proximal dendrites, with larger sizes, more responsive to afferent stimulation, with a larger number of calcium permeable channels, with stores more full of calcium. If the scope of this paper is to characterize LTP_pre_/LTD_pre_, it would be then more appropriate to use postsynaptic calcium transients to obtain a careful time-dependent comparison of the effects of the different stimulation paradigms and drug manipulation-dependent comparison (see comment below) and use electrophysiology to confirm optical data. Average results from LTP/LTD optical experiments should be presented as average time courses across many synapses, cells and conditions to illustrate time-dependent changes and to compare the effects of the different stimulation paradigms and drug manipulations. The synaptic selection method and the location and morphological characteristics of the investigated synapses in these experiments should be better explained in the Materials and methods section discussing potential caveats.

Unfortunately, the authors extensively use a change in pair-pulse facilitation as a reliable way to demonstrate changes occurring in the presynapse (Figure 1, Figure 3, Figure 3—figure supplement 1, Figure 3—figure supplement 2, Figure 5—figure supplement 1, Figure 5—figure supplement 2, Figure 7, Figure 7—figure supplement 1). The authors state: "[…]robust and reliable LTP which had a presynaptic component of expression as assessed by a decrease in the pair pulse ratio". This is a weak and very indirect argument that could lead to false conclusions. The mechanisms behind PPF in the hippocampus are still debated, the most popular hypothesis explains presynaptic facilitation by the enhanced recruitment of quanta from calcium left inside the presynapse, following the first stimulus in the pair. Even assuming that this hypothesis was true (some recent results on presynaptic proteins speak against it, for example those on the triple KO for synapsins), a drop in the pair-pulse ratio can be seen only if the site of action of LTP_pre_ is shared out with PPF. This is unlikely because the synaptic release process involves many passages, and if LTP_pre_ were to act upstream or downstream the PPF-locus, the pair-pulse ratio would not be changed (LTP_pre_ would have a proportional effect on the first and second response). More importantly for this study, a change in PPF is not a sufficient indication for a presynaptic change. Some post-synaptic processes (for example a change in local dendritic potential, a fast desensitization of post-synaptic receptors) do affect PPF. Furthermore, a specific problem with cultured slices experiments is the high incidence of polysynaptic connections which generate reverberant activity, often contaminating the PPF observation window. These caveats should be discussed in the paper.

2) Calcium imaging data are from small N (N = 12 spines; N = 5 spines; N = 3 spines (subsection “Glutamate photolysis inhibits LTP_pre_ and promotes LTD_pre_”, third paragraph)) – the authors have to increase the sample size. It is also unclear if these values refer to spines from one or from a set of neurons. The number of independent experiments and the number of spines analyzed for each condition should be clearly indicated in each case. If the N is from just one or few experiments there will be a strong dependency of measured data, hence pooled observations would not be independent and there won't be equal variation across different experimental manipulations. In this respect it would be important to explain how the most adequate sample size was set to reach a meaningful statistical power.

3) The authors' claim that nitric oxide is *the* retrograde messenger at CA3-CA1 synapses is too strong. This claim is based on PTIO and L-NAME pharmacological experiments, which are not entirely definitive. The donor uncaging experiment only proves that nitric oxide is sufficient for LTP_pre_. Given that several other candidates for retrograde messenger have been described, we recommend that the conclusion be softened (i.e., “LTP_pre_ requires nitric oxide signalling”) and other candidates be discussed. Moreover, the authors should provide time courses of DAF-FM and DAQ fluorescence increases to interpret the kinetics of NO release from postsynaptic cells.

4) The evidence for functional presynaptic NMDARs at CA3-CA1 synapses heavily relies on pharmacology. The experiments presented in the paper don't exclude the involvement of NNMDARs expressed by astrocytes. The authors may use the Cre-Lox system to remove GluN1 from CA3 pyramidal neurons or use a knockdown strategy. If not, alternative possibilities should be discussed in the paper.

---

## [Author Response]

Despite the importance of all topics touched in this work and their impressive number, the authors need to strengthen the direct evidence of the LTP_pre_ in the observed phenomena. Electrophysiological tools do not allow for unequivocal proof of LTP_pre_ / LTD_pre_ expression and modulation. Thus, the reviewers expect that the authors strengthen their data using optical tools to monitor CaT or synaptic vesicle exocytosis to support the major conclusions of the paper.Please consider the detailed concerns below and write back with your plans to address these experimentally and an estimate of the time it will take to do so. We will share your response with the Board and reviewers who will then confer to issue their thoughts on your proposed experiments.

We would like to thank the reviewers for their thoughtful and constructive comments on our work. We found the reviewers’ comments very helpful in guiding our experimental plans and revisions for the purposes of presenting a stronger case for our findings.

We have now conducted all of the revisions and experiments we had originally proposed to address the reviewers’ concerns. This includes strengthening our evidence for presynaptic changes by 1) increasing our sample sizes and 2) using time lapse Ca^2+^ imaging and paired pulse ratios (PPR) to more extensively characterize presynaptic changes across time. Perhaps most significantly, we have done these time lapse experiments in conditional GluN1 knockout mice to confirm the roles of pre- and postsynaptic NMDARs in LTP_pre_ and LTD_pre_ induction. We feel that the newly incorporated experiments have greatly strengthened our findings and provide further support for our proposed framework of presynaptic plasticity.

We address the reviewers’ comments below.

Essential revisions:1) Why didn't the authors base the entire experimental design on the optical methodology they have developed in the past (Emptage et al., 2003; Emptage, Bliss and Fine, 1999)? Imaging the number of successful postsynaptic calcium transients (assuming that the responses to single quanta of glutamate are indeed reliably detected) would focus on presynaptic changes thus being much more direct and rewarding. This approach was used here but just as a confirmation of the electrophysiological results (Figure 2, Figure 4, Figure 5, Figure 6). Unfortunately, results from recordings of EPSPs and from optical experiments are not superimposable. Intracellular recordings of synaptic potentials sample large populations of synapses, a likely mix of different presynaptic and postsynaptic phenotypes and forms of plasticity. On the contrary, postsynaptic calcium transients sample a small subset of these, presumably biasing the selection toward spiny synapses located on proximal dendrites, with larger sizes, more responsive to afferent stimulation, with a larger number of calcium permeable channels, with stores more full of calcium.

The study uses both Ca^2+^ imaging and electrophysiology to support its conclusions. The reason for using both techniques is that we recognise that each has its own advantages and disadvantages. We therefore believe that both techniques combined provide a much stronger case of support for our results, and are more likely to appeal to a wider audience, including scientists from both imaging and electrophysiology communities.

On the basis of the reviewers’ comments, we have now discussed the caveats and strengths of both techniques in the Discussion. The relevant portion of the discussion is pasted below:

“Discussion – Experimental techniques

Our study is robust because we examine presynaptic plasticity under a diverse range of experimental conditions. […] Both of these techniques, therefore, present valid means of measuring presynaptic efficacy, at least under the experimental conditions used in this study.”

If the scope of this paper is to characterize LTP_pre_/LTD_pre_, it would be then more appropriate to use postsynaptic calcium transients to obtain a careful time-dependent comparison of the effects of the different stimulation paradigms and drug manipulation-dependent comparison (see comment below) and use electrophysiology to confirm optical data. Average results from LTP/LTD optical experiments should be presented as average time courses across many synapses, cells and conditions to illustrate time-dependent changes and to compare the effects of the different stimulation paradigms and drug manipulations.

The idea of using Ca^2+^ imaging to assess Pr over time is an excellent suggestion. As proposed, we assessed P_r_ at a number of time points (baseline, +5, +15, +30, +45 minutes) following plasticity induction. We did this in slices in which GluN1, the obligatory NMDAR subunit, was knocked out presynaptically (i.e. in CA3 neurons). Consistent with our previous results, we found that knockout of presynaptic NMDARs augmented increases in Pr induced by paired stimulation and abolished decreases in Pr induced by unpaired stimulation (Figure 8). Thus, changes in P_r_ were always greater than control when presynaptic NMDARs were knocked out, regardless of whether we were delivering paired our unpaired stimulation. These findings confirm that presynaptic NMDAR activation serves to promote decreases in P_r_.

Reassuringly, we found that increases in Pr following paired stimulation and decreases in Pr following unpaired stimulation grew over time (Figure 8), consistent with the initial observation by Bayazitov et al. (2007) that presynaptic changes evolve over time. We make mention of this observation in the paper. Moreover, the effects of presynaptic NMDARs on Pr were observable throughout the post-induction period, starting from 5 minutes. This suggests that presynaptic NMDARs are unlikely to dynamically alter the evolving expression of presynaptic plasticity.

As suggested by the reviewers, we also complemented our imaging findings with electrophysiological data (Figure 7) in these slices. We examined both the EPSP and PPR over time following paired and unpaired stimulation (Figure 7). As can be seen, changes in EPSP and PPR nicely mirror one another. When PPR changes from electrophysiological experiments were plotted alongside changes in Pr obtained from imaging experiments (Figure 8), we found nice correspondence between the two measures. Such findings help to validate both PPR and Ca^2+^ imaging as techniques for assessing presynaptic efficacy.

In electrophysiological experiments, we additionally examined the effects of knocking out postsynaptic NMDARs (i.e. in CA1 neurons) (Figure 7). Under these conditions, we were still able to induce LTP_pre_ and LTD_pre_. Consistent with our previous findings, this suggest that postsynaptic NMDARs are not strictly required for the induction of presynaptic plasticity.

The synaptic selection method and the location and morphological characteristics of the investigated synapses in these experiments should be better explained in the Materials and methods section discussing potential caveats.

We have discussed the method of synapse selection more clearly in the Materials and methods, and have stated that Ca^2+^ imaging biases us to sampling mushroom spines that have larger Ca^2+^ transients. We also mention this caveat in the Discussion section. The relevant Materials and methods section reads:

“To find a synapse responsive to axonal stimulation, axons were stimulated with pairs of stimuli (2 pulses 70 ms apart) to increase the chances of eliciting a Ca^2+^ response. […] Synapse selection, however, was invariably biased in favour of mushroom spines, with head diameters ranging from 0.3-1.0 µm, as these synapses were clearly visible and produced larger Ca^2+^ transients.”

Unfortunately, the authors extensively use a change in pair-pulse facilitation as a reliable way to demonstrate changes occurring in the presynapse (Figure 1, Figure 3, Figure 3—figure supplement 1, Figure 3—figure supplement 2, Figure 5—figure supplement 1, Figure 5—figure supplement 2, Figure 7, Figure 7—figure supplement 1). The authors state: "[…]robust and reliable LTP which had a presynaptic component of expression as assessed by a decrease in the pair pulse ratio". This is a weak and very indirect argument that could lead to false conclusions. The mechanisms behind PPF in the hippocampus are still debated, the most popular hypothesis explains presynaptic facilitation by the enhanced recruitment of quanta from calcium left inside the presynapse, following the first stimulus in the pair. Even assuming that this hypothesis was true (some recent results on presynaptic proteins speak against it, for example those on the triple KO for synapsins), a drop in the pair-pulse ratio can be seen only if the site of action of LTP_pre_ is shared out with PPF. This is unlikely because the synaptic release process involves many passages, and if LTP_pre_ were to act upstream or downstream the PPF-locus, the pair-pulse ratio would not be changed (LTP_pre_ would have a proportional effect on the first and second response). More importantly for this study, a change in PPF is not a sufficient indication for a presynaptic change. Some post-synaptic processes (for example a change in local dendritic potential, a fast desensitization of post-synaptic receptors[…]) do affect PPF. Furthermore, a specific problem with cultured slices experiments is the high incidence of polysynaptic connections which generate reverberant activity, often contaminating the PPF observation window. These caveats should be discussed in the paper.

We do appreciate the reviewers concerns and we do agree that the use of paired pulse ratio (PPR) is not without issue. In the Discussion we have stated that PPR is an indirect measure of presynaptic efficacy that could be potentially confounded by postsynaptic changes. However, we should express the following:

1) The strength of our paper lies in using both PPR and Ca^2+^ imaging to investigate presynaptic plasticity. We have consistently found good agreement between both of these measures across our study. Perhaps this is best exemplified with the newly included experiments, in which time lapse PPR recordings and Ca^2+^ imaging showed very similar trends (Figure 8) following both LTP and LTD induction. Thus, although the exact mechanisms underlying PPR may be debated, at least within our experimental conditions, PPR functions as a good readout for presynaptic changes.

2) We have previously reported that at single synapses, PPR decreases after LTP. Using Ca^2+^ imaging, Emptage et al. (2003) assessed Pr during the first and second pulse of stimulation during paired pulse stimulation (ISI 70ms). They took the ratio of the two [Pr(2^nd^ pulse)/Pr(1^st^ pulse)] to calculate PPR and found that individual synapses show paired pulse facilitation, and that the PPR at these synapses decreases after LTP. Similarly, in our current study, in response to the reviewers’ comments we looked at PPR changes at single synapses in one set of Ca^2+^ imaging data. We found that single synapse PPR decreased following LTP_pre_ induction (ΔPPR = -0.42 ± 0.12; n = 12; p<0.01), and increased following LTD_pre_ induction (ΔPPR = +0.97 ± 0.24; n = 9; p<0.01). Again, while the mechanisms underlying PPR and its associated changes can be debated, we find that at the level of single synapses, changes in PPR measured with Ca^2+^ imaging are consistent with changes in PPR recorded more globally across synapses using electrophysiology. This suggests that bonafide changes in PPR do occur in response to LTP and LTD induction, and that changes in PPR recorded electrophysiologically are a good proxy for assessing presynaptic changes, at least in our experimental conditions.

3) Although postsynaptic effects, namely receptor desensitization, can confound PPR measures, they do not affect PPR measurements at the interpulse interval (70 ms) used in our study (Yang and Calakos, 2013).

4) Although polysynaptic activity can be problematic for PPR analysis, reverberant activity was minimal in our cultured slice (see representative PPR traces in Figure 1—figure supplement 1). Moreover, we always quantified EPSPs using their initial slope (first 2-3 ms) rather than their amplitude to prevent polysynaptic activity from confounding our analysis. Furthermore, from our results it is evident that results from PPR analyses in cultured slices yielded similar results to that obtained in acute slices, in which polysynaptic activity is less of a concern (see for example, Figure 3 and Figure 3—figure supplement 2, and Figure 6—figure supplement 4 and Figure 6—figure supplement 5).

2) Calcium imaging data are from small N (N = 12 spines; N = 5 spines; N = 3 spines (subsection “Glutamate photolysis inhibits LTP_pre_ and promotes LTD_pre_”, third paragraph)) – the authors have to increase the sample size. It is also unclear if these values refer to spines from one or from a set of neurons. The number of independent experiments and the number of spines analyzed for each condition should be clearly indicated in each case. If the N is from just one or few experiments there will be a strong dependency of measured data, hence pooled observations would not be independent and there won't be equal variation across different experimental manipulations. In this respect it would be important to explain how the most adequate sample size was set to reach a meaningful statistical power.

Only one spine was imaged per cell per experiment; we apologize for not making this clearer; we have updated our Methods section accordingly.

As proposed, we have now increased our N’s such that all imaging data sets have N > 8. In total, 9 experimental conditions were revisited for these purposes. All data sets have been updated in the figures and in the main text. None of our conclusions have changed. We should also stress that, because our effect sizes are large, we are able to achieve a statistical power of > 90% across experimental conditions with our sample sizes.

For the reviewers’ interest, our sample sizes are low because Ca^2+^ imaging, at least within the hippocampus, does not allow for a large number of synapses to be analysed in parallel. At least in our hands, the technique is generally quite difficult, especially in the context of plasticity experiments because of the time (>35 minutes) required to maintain electrical control over the imaged synapse. Even after finding a synapse responsive to stimulation, many experiments fail owing to even slight drift in the stimulating electrode during the time course of the experiment (which is generally tolerable when recording EPSPs, but not when recording Ca^2+^ transients), resulting in the loss of presynaptic control over the imaged synapse. This is typically why the N’s are low in plasticity experiments using Ca^2+^ imaging, especially when measurements need to be obtained at the same spine after a period of time. For example, in our previous studies (Emptage et al., 2013, Neuron and Ward et al., 2006, Neuron) we use an N of 4-13 and 6-14 respectively, depending on the difficulty of the experiment. A similar study of presynaptic plasticity by Alan Fine’s group (Enoki et al., 2009, Neuron) used an N of 5-8 throughout their study. A paper from the Sabatini lab (Kwon and Sabatini, 2011, Science) used an N of 5-11 in their study in an experiment which required assessing Pr of newly formed spines after several minutes.

3) The authors' claim that nitric oxide is the retrograde messenger at CA3-CA1 synapses is too strong. This claim is based on PTIO and L-NAME pharmacological experiments, which are not entirely definitive. The donor uncaging experiment only proves that nitric oxide is sufficient for LTP_pre_. Given that several other candidates for retrograde messenger have been described, we recommend that the conclusion be softened (i.e., “LTP_pre_ requires nitric oxide signalling”) and other candidates be discussed.

We have now softened the claims made in the manuscript, by referring to NO as

“one promising candidate” for retrograde signalling. We have also briefly discussed other retrograde signalling possibilities in the Discussion. The relevant section now reads:

“While we provide evidence in support of NO as a retrograde signal in LTP_pre_, it may not be the only retrograde signal involved. Indeed, neurotrophic factors, trans-synaptic signals, as well as contact-dependent processes are all known to regulate P_r_ (Regehr et al., 2009); whether such signals play a role in LTP_pre_ induction remains to be elucidated.”

Moreover, the authors should provide time courses of DAF-FM and DAQ fluorescence increases to interpret the kinetics of NO release from postsynaptic cells.

We are unable to conduct time-lapse imaging of DAF-FM and DAQ fluorescence. Unfortunately, both dyes are highly susceptible to photobleaching, and given that the absolute fluorescent signal change in response to physiological levels of NO release is quite small, photobleaching is sufficient to destroy the signal in our hands. Consequently, in our experiments we restricted imaging to two time points: baseline and post-stimulation.

4) The evidence for functional presynaptic NMDARs at CA3-CA1 synapses heavily relies on pharmacology. The experiments presented in the paper don't exclude the involvement of NNMDARs expressed by astrocytes. The authors may use the Cre-Lox system to remove GluN1 from CA3 pyramidal neurons or use a knockdown strategy. If not, alternative possibilities should be discussed in the paper.

As proposed in our revision plan, and discussed in Essential revision 1, we have now used the Cre-Lox system, as recommended by the referees, to more directly explore the role of pre- and postsynaptic NMDARs in presynaptic plasticity (Figure 7, Figure 8). We find, consistent with our pharmacological results, that presynaptic NMDAR activation decreases P_r_ during both LTP_pre_ induction and LTD_pre_ induction.

Postsynaptic NMDARs, by contrast, have no effect on activity-dependent changes in Pr under our experimental conditions.